# Risk of yellow fever virus transmission in the Asia-Pacific region

Lucy de Guilhem de Lataillade[1], Marie Vazeille[1], Thomas Obadia[2,3], Yoann Madec [4], Laurence Mousson[1], Basile Kamgang [5], Chun-Hong Chen [6], Anna-Bella Failloux [1✉] & Pei-Shi Yen [1✉]

Historically endemic to Sub-Saharan Africa and South America, yellow fever is absent from the Asia-Pacific region. Yellow fever virus (YFV) is mainly transmitted by the anthropophilic *Aedes* mosquitoes whose distribution encompasses a large belt of tropical and sub tropical regions. Increasing exchanges between Africa and Asia have caused imported YFV incidents in non-endemic areas, which are threatening Asia with a new viral emergence. Here, using experimental infections of field-collected mosquitoes, we show that Asian-Pacific *Aedes* mosquitoes are competent vectors for YFV. We observe that *Aedes aegypti* populations from Singapore, Taiwan, Thailand, and New Caledonia are capable of transmitting YFV 14 days after oral infections, with a number of viral particles excreted from saliva reaching up to 23,000 viral particles. These findings represent the most comprehensive assessment of vector competence and show that *Ae. aegypti* mosquitoes from the Asia-Pacific region are highly competent to YFV, corroborating that vector populations are seemingly not a brake to the emergence of yellow fever in the region.

[1] Arboviruses and Insect Vectors Unit, Institut Pasteur, Paris, France. [2] Bioinformatics and Biostatistics Hub, Institut Pasteur, USR 3756, CNRS, Paris, France. [3] Malaria Unit: Parasites and Hosts, Institut Pasteur, Paris, France. [4] Emerging Diseases Epidemiology Unit, Institut Pasteur, Paris, France. [5] Department of Medical Entomology, Centre for Research in Infectious Diseases, Yaoundé, Cameroon. [6] National Health Research Institutes, Institute of Infectious Diseases and Vaccinology, Miaoli, Taiwan. ✉email: anna-bella.failloux@pasteur.fr; pei-shi.yen@pasteur.fr

In 2016, the return from Angola of 11 yellow fever (YF)-infected workers to China posed the threat of a YF epidemic in Asia never before seen[1]. Increasing volumes of trade and travels between China and Africa increase the risk of disease introductions. Yellow fever virus (YFV), endemic to Africa and South America, has so far remained absent in Asia. The reasons explaining this absence (e.g., transmission barrier resulting from low compatibility between mosquito and virus genotypes[2,3], limited duration and low viraemia in humans, absence of a syl-vatic cycle[4,5], competition with well-established flaviviruses as dengue and Japanese encephalitis viruses[6]) are still poorly explored, making the possibility of an epidemic unpredictable.

Similar to other flaviviruses, the common symptoms of YF are fever, headache, muscle aches, nausea, and vomiting, however, the in-hospital case fatality rate (CFR) could drama-tically reach 67%[7,8], giving this disease a particular interest for public health. Traced back to around 3000 years ago, YF was mainly encountered in Africa where it was isolated in 1927 in Ghana[2]. YFV was transported via ships sailing from West Africa to the West Indies during the slave trade. Massive and recurrent transports of goods also brought competent vectors such as the mosquito Aedes aegypti contributing to initiate YFV transmission cycles in ships and later, on land at their desti-nation. The in-depth understanding of YFV transmission cycle in the early 1900s[9] permitted to implement successful vector control strategies since 1916[10] and to develop the YFV 17D vaccine in 1936[11]. However, YFV still causes an estimated 51,000–380,000 annual severe cases, of which 19,000–180,000 are fatal in Africa[12]. Insecticide-resistance of mosquito popu-lations, as well as a supply shortage, distribution, and uptake of YFV vaccines, are among the main causes of this current burden[13].

To transmit an arbovirus such as YFV, the mosquito should acquire the virus by ingesting viremic blood from an infected host, the virus enters into the midgut epithelial cells and repli-cates. After a few days of incubation, the virus should pass through the midgut basal lamina and disseminate into the hemocele, then it infects the salivary glands for transmission to the vertebrate host[14]. Parameters such as midgut infection, viral dissemination in hemocele, and transmission through infectious saliva are used to determine mosquito vector competence, which is an indicator of transmission risk[15]. In Africa and South America, YFV is primarily transmitted in a forest cycle between non-human primates (NHP) and zoophilic mosquitoes (Aedes in Africa and Haemagogus/Sabethes in South America). The urban cycle of YFV involves mainly the mosquito Ae. aegypti in both Africa and South America[16].

The mosquito vectors Ae. aegypti and Aedes albopictus, are present in 154 countries putting nearly half of the world population at risk of YFV transmission. Ecological disturbances induced by urban habitats contribute to the proliferation of Ae. aegypti, supplanting Ae. albopictus in urban areas in Asia[17]. Aedes spp. mosquitoes are vectors of chikungunya, dengue, and Zika viruses in East and South-East Asia, which serve as a suitable environment for YFV. Increasing exchanges between Asia and Africa has raised the number of passengers between Asia and YF-endemic countries[18–20]. Notable increase of travels between countries with different capacities to detect and con-trol infectious diseases (e.g., growth of tourism in emerging countries) can facilitate the geographic spread of vector-borne diseases[20,21].

Of greater concern was the report of YFV laboratory-confirmed cases among Chinese travelers returning to Asia after a stay in Angola during the 2015–2016 YF outbreak[3], threa-tening billions of immunologically naive populations in Asia living in close vicinity of Ae. aegypti and Ae. albopictus

mosquitoes[1]. Africa receives a large number of Chinese workers who are usually unvaccinated against YFV, increasing the risk of importing YF in Asia[22]. The combination of repeated introductions of viraemic travelers and immunologically naive local population in an environment suitable to transmission accentuates the risk of YF emergence in Asia. Although the vector competence for YFV of mosquitoes in Africa, South America, and Caribbean regions, has been investigated[23,24], only limited information for Asian-Pacific mosquitoes could be found to measure the possible risk of YFV transmission in this region[25,26]. Investigating the vector competence for YFV of mosquitoes in the Asia-Pacific region is essential to assess the potential threat of YFV transmission in a region where YF outbreaks have never been reported[27]. Here, we show the vector competence of 18 populations of Ae. aegypti and Ae. albopictus from the Asia-Pacific region. We demonstrate that (i) Ae. aegypti mosquitoes from the Asia-Pacific region are more sus-ceptible to the West-African genotype of YFV than Ae. albo-pictus, (ii) mosquitoes from Singapore, Taiwan, Thailand, and New Caledonia are capable of transmitting YFV at 14 days post-infection, and (iii) Ae. aegypti mosquitoes excrete up to 23,000 viral particles in saliva, suggesting that YFV could be transmitted through the saliva of infected Ae. aegypti in laboratory conditions.

## Results

**Aedes aegypti mosquitoes are highly competent to YFV infec-tion.** Aedes aegypti populations from the Asia-Pacific region were used in experimental infections to evaluate different components of the vector competence at 14 and 21 days post-infection (dpi).

At 14 dpi, infection rate (IR) ranged from 41.7% (CAMB, Cambodia) to 95.8% (TRUNG, Vietnam; CSP, Thailand; TAINAN, Taiwan; NOUMEA, New Caledonia) and were significantly different when comparing all 12 populations (Fish-er's exact test: $P < 10^{-4}$), 10 Asian populations ($P < 10^{-4}$) and the two populations from the Pacific region ($P = 0.02$) (Fig. 1a). Dissemination rate (DR) ranged from 42.8% (FENG, Taiwan) to 86.9% (CSP, Thailand; NOUMEA, New Caledonia), with some populations presenting higher DR (Fisher's exact test: $P = 0.06$); the 10 Asian populations presented similar DR ($P = 0.13$) while the two populations from the Pacific region presented signifi-cantly different DR ($P = 0.04$) (Fig. 1b). Based on transmission rate (TR), seven among 12 populations did not excrete virus in saliva. For the other five populations (CSP, SING, ANNAN, FENG, NOUMEA), TR ranged from 12.5% (SING, Singapore) to 45% (CSP, Thailand), and was significantly different (Fisher's exact test: $P < 10^{-4}$) (Fig. 1c).

To test whether a longer incubation time of mosquitoes might improve the vector competence, we used the same protocol to assess IR, DR, and TR at 21 dpi. IR reached 100% in four populations (BLX, Laos; CSP, Thailand; SING, Singapore; NOUMEA, New Caledonia), and remained significantly different between the 12 populations (Fisher's exact test: $P < 10^{-4}$) (Fig. 1d). DR ranged from 47.6% (NANZI, Taiwan) to 95.8% (CSP, Thailand) and differed between populations (Fisher's exact test: $P < 10^{-4}$) (Fig. 1e). However, TR ranged from 10% (VIET, Vietnam) to 56.5% (CSP, Thailand), but no significant difference between the 12 populations was evidenced (Fisher's exact test: $P = 0.10$) (Fig. 1f). Collectively, these results show that all Ae. aegypti mosquitoes examined in this study are competent vectors of YFV with 42% of populations (5/12) able to transmit at 14 dpi and all (12/12) at 21 dpi.

**Aedes albopictus mosquitoes are less competent to YFV than Ae. aegypti.** To examine whether Ae. albopictus native to Asia can

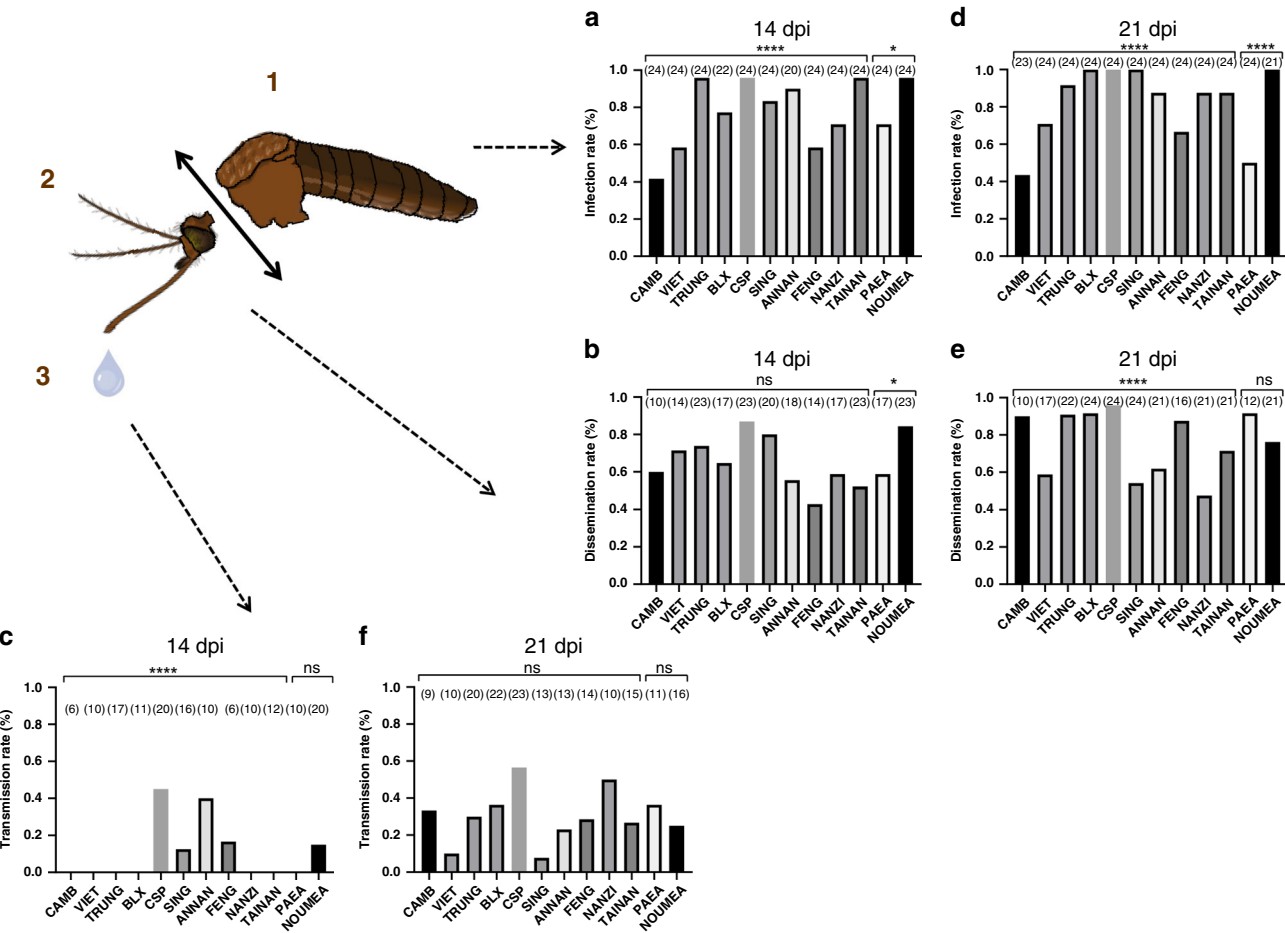

**Fig. 1 Vector competence of 12 *Aedes aegypti* populations assessed 14 and 21 days after an infectious blood meal containing $10^7$ ffu/mL of YFV (West-African genotype).** Batches of 20–24 mosquitoes were examined in each population for viral infection (**a**, **d**), dissemination (**b**, **e**), and transmission (**c**, **f**) by estimating respectively the proportion of mosquitoes with infected bodies (1), head (2), and saliva (3). Titrations were performed on C6/36 cells in 96-well plates. Viral particles were detected by FFA using a primary anti-YFV antibody and a secondary fluorescent-conjugated antibody. Infection rate (IR) refers to the percentage of mosquitoes having an infected body among blood-fed mosquitoes. Dissemination rate (DR) is the percentage of mosquitoes with an infected head (containing viral particles having disseminated in the general cavity after crossing successfully the midgut) among mosquitoes with an infected body. Transmission rate (TR) corresponds to the percentage of mosquitoes with infectious saliva (viral particles having successively crossed the two anatomical barriers, midgut and salivary glands) among mosquitoes with infected head. Stars indicate statistical significance of comparisons by Fisher's exact test (two-sided test; *$P \le 0.05$, ****$P \le 0.0001$). **a** ****$P \le 0.0001$, *$P = 0.02$; **b** *P0.042; **c** ****$P \le 0.0001$; **d** ****$P \le 0.0001$; **e** ****$P \le 0.0001$. ns (non-significant) indicates a lack of statistical significance ($P > 0.05$). In brackets are the numbers of mosquitoes tested. dpi days post-infection. Source data are provided in Supplementary Data 1 file.

sustain a local transmission of YFV, vector competence indices, IR, DR, and TR were calculated for six populations at 14 and 21 dpi.

At 14 dpi, IR ranged from 4.2% (THAI, Thailand) to 62.5% (FOSHAN, China), and significantly differed between populations (Fisher's exact test: $P < 10^{-4}$) (Fig. 2a). DR ranged from 0% (THAI, Thailand) to 85.7% (LINGYA, Taiwan), but no significant difference was evidenced between populations (Fisher's exact test: $P = 0.41$) (Fig. 2b). The transmission was only observed for FOSHAN (TR = 22.2%) (Fig. 2c).

At 21 dpi, IR ranged from 8.3% (THAI, Thailand) to 54.2% (FOSHAN, China) (Fisher's exact test: $P = 0.003$) (Fig. 2d). DR ranged from 0% (XKH, Laos and THAI, Thailand) to 100% (LINGYA (Taiwan)) (Fisher's exact test: $P = 0.04$) (Fig. 2e), and TR to 66.7% (LINGYA, Taiwan) (Fig. 2f). Four *Ae. albopictus* populations (YYG, Japan; XKH, Laos; THAI, Thailand; PMNI, Brazil) were not able to transmit at both 14 and 21 dpi. These results indicate that *Ae. albopictus* populations are less competent to disseminate and transmit YFV than *Ae. aegypti* (Supplementary Figs. 1 and 2).

**Higher loads of viral particles excreted in the saliva of *Ae. aegypti* than *Ae. albopictus*.** To study whether *Ae. aegypti* delivered a higher load of viruses in saliva than *Ae. albopictus*, we collected individual mosquito saliva that was titrated. We observe that among the five populations able to transmit at 14 dpi, the number of viral particles varied from $10^{1.6 \pm 1.5}$ (NOUMEA, New Caledonia) to $10^3$ (FENG, Taiwan) (Fig. 3a). At 21 dpi, all 12 populations deliver viral particles in saliva ranging from 5 (VIET, Vietnam) to $10^{3.7 \pm 4}$ (NANZI, Taiwan: min-max: 10–23,000) (Fig. 3b). Comparatively, *Ae. albopictus* mosquitoes were able to deliver $10^{1.7 \pm 1.7}$ viral particles (FOSHAN, China) at 14 dpi (Fig. 3c) and $10^{2.2 \pm 1.4}$ viral particles (LINGYA, Taiwan; min-max: 133–167) at 21 dpi (Fig. 3d).

**Lower dissemination of YFV in *Ae. aegypti* from the Asia-Pacific region compared to African mosquitoes.** To evaluate whether higher viral loads in the body and head of mosquitoes could increase the chance for virus transmission through saliva, viral particles in the body, head, and saliva were estimated only

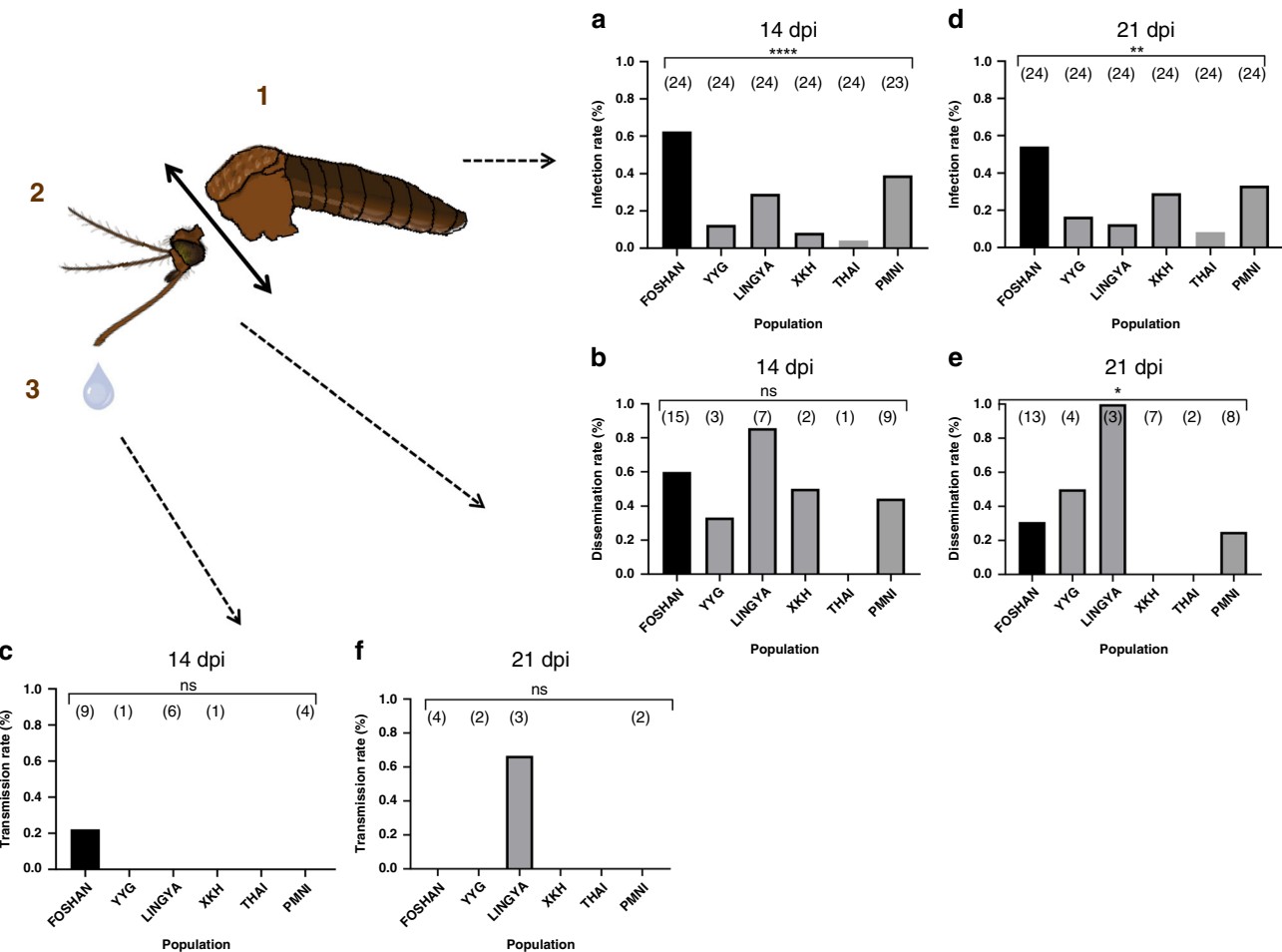

**Fig. 2 Vector competence of 6 *Aedes albopictus* populations assessed 14 and 21 days after an infectious blood meal containing 10⁷ ffu/mL of YFV (West-African genotype).** Batches of mosquitoes were examined in each population for viral infection (**a**, **d**), dissemination (**b**, **e**), and transmission (**c**, **f**) by estimating respectively the proportion of mosquitoes with infected bodies (1), head (2), and saliva (3). Infection rate (IR) refers to the percentage of mosquitoes having an infected body among blood-fed mosquitoes. Dissemination rate (DR) is the percentage of mosquitoes with an infected head (containing viral particles having disseminated in the general cavity after crossing successfully the midgut) among mosquitoes with an infected body. Transmission rate (TR) corresponds to the percentage of mosquitoes with infectious saliva (viral particles having successively crossed the two anatomical barriers, midgut and salivary glands) among mosquitoes with infected head. Stars indicate statistical significance of comparisons by Fisher's exact test (two-sided test; $*P \leq 0.05$, $**P \leq 0.01$, $****P \leq 0.0001$). **a** $****P \leq 0.0001$; **d** $**P = 0.003$; **e** $*P = 0.038$. ns (non-significant) indicates a lack of statistical significance ($P > 0.05$). In brackets are the numbers of mosquitoes tested. dpi days post-infection. Source data are provided in Supplementary Data 1 file.

for mosquitoes capable of viral transmission (Supplementary Fig. 3). Viral loads in the body (Fig. 4a) and saliva (Fig. 4c) were not significantly different (Kruskal–Wallis: $P > 0.05$), while viral loads in the head were significantly higher (Fig. 4b) in mosquitoes from Africa ($10^{4.6\pm3.7}$) compared to Asia ($10^{3.9\pm3.9}$) and Pacific ($10^{3.7\pm3.6}$) regions (Kruskal–Wallis test: $P > 0.05$).

Viral loads in the body were significantly correlated with viral loads in the head ($\rho = 0.31$, $P = 0.012$) (Supplementary Fig. 4a). However, no correlation was detected between viral loads in body and saliva ($\rho = 0.22$, $P = 0.11$; Supplementary Fig. 4b), or between virals loads in head and saliva ($\rho = 0.04$, $P = 0.77$; Supplementary Fig. 4c). To investigate the difference in terms of viral loads in body, head, and saliva between mosquitoes from different geographic origins, we used a linear regression model. To identify the main factor conditioning the correlation between viral loads in body, head, and saliva, we used a logistic regression model. The analysis corroborated that compared to mosquitoes from Africa used as the reference, *Ae. aegypti* from Asia presented a lower viral load in the head (the level is $-0.73$ log lower in mean in Asian than in African mosquitoes) as for mosquitoes from the Pacific region ($-0.82$ in mean) (Table 1, $P = 0.01$).

When analyzing viral loads in body and saliva, no significant difference was found between mosquitoes from Africa, Asia, and the Pacific region (Table 1, respectively, $P = 0.11$ and $P = 0.54$).

Taken altogether, these results indicate that compared to mosquitoes from Africa, *Ae. aegypti* mosquitoes from the Asia-Pacific region hosted significantly lower viral particles in the head but presented similar viral loads in body and saliva, suggesting that only viral dissemination distinguishes *Ae. aegypti* mosquitoes from the three continents.

**Ability of different *Aedes aegypti* populations to transmit YFV.** To determine the risk of mosquito-mediated YFV transmission at each location, we used transmission efficiencies (Supplementary Fig. 1) and probabilities of vector occurrence (data from Kraemer et al.[28]). When considering only *Ae. aegypti* from Asia, regions where CSP (Thailand), TRUNG (Vietnam) and NANZI (Taiwan) populations are located, presented a higher transmission risk of YFV (CSP: 54% [32.8–74.4%], TRUNG: 25% [9.8–48.7%], NANZI: 21% [7.1–42.2%]). In these regions, mosquito occurrence is predicted to be high and overall constant within a 5 km radius, allowing for competent vectors to place immunological naive

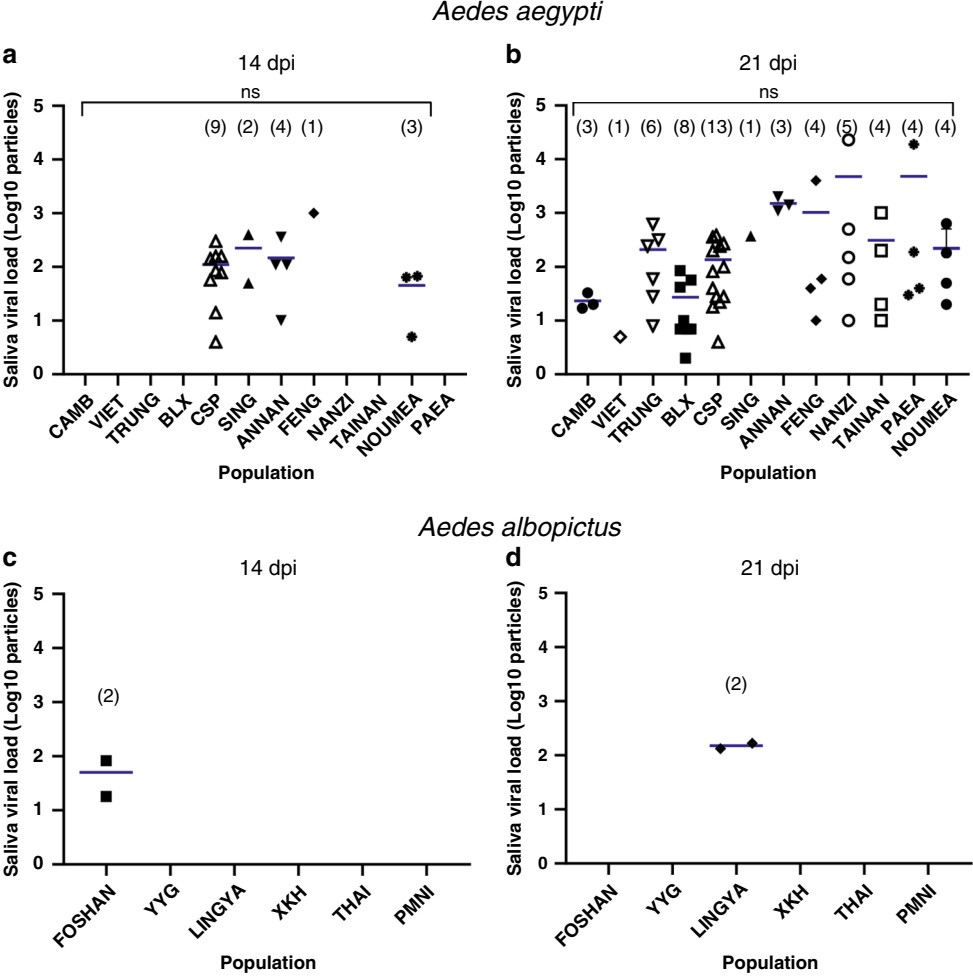

*Aedes aegypti*

*Aedes albopictus*

**Fig. 3 Viral loads measured in individual mosquito saliva at 14 and 21 days after an infectious blood meal with West-African YFV. a, b** Saliva viral loads of *Aedes aegpti* at 14 and 21 dpi; **c, d** saliva viral loads of *Aedes albopictus* at 14 and 21 dpi. Saliva was collected for 30 min using the forced salivation technique by removing legs and wings and inserting a tip containing FBS in mosquito proboscis. Salivas were titrated on C3/36 cells and the numbers of viral particles are expressed in ffu/saliva. ns (non-significant) indicates the lack of statistical significance for comparisons using the Kruskal–Wallis test (two-sided test; $P > 0.05$). Bars indicate the mean. In brackets are the numbers of mosquitoes tested. dpi days post-infection. Source data are provided in Supplementary Data 1 file.

populations (humans and natural reservoirs) at risk of YFV infection (Fig. 5).

## Discussion

To our knowledge, only two studies have been published on vector competence for YFV of mosquitoes from the Asia-Pacific region[25,26]. In our study, we examine 18 mosquito populations and find that *Ae. aegypti* populations from the Asia-Pacific region are more competent to transmit YFV than *Ae. albopictus* from the same geographical area. Compared to *Ae. aegypti* from YFV-endemic regions in Africa, mosquitoes from Singapore, Taiwan, Thailand, and New Caledonia presented the highest potential to transmit YFV; the risk of transmission to human populations is high. Based on these results, we cannot exclude the possibility of a YF epidemic occurring in the Asia-Pacific region where *Ae. aegypti* is well-established.

A previous study using Asian *Ae. aegypti* populations showed that in laboratory conditions, *Ae. aegypti* from Laos (Bolikhamsai province) were able to transmit YFV at least 14 days after exposure to YFV S-79 strain[25]. Conversely, when infected with the American genotype 1 of YFV (strain 74018, from Brazil), *Ae. aegypti* from Cambodia (Phnom Penh) and Vietnam (Ho Chi Minh city) were found to be susceptible to YFV[26] with however

lower dissemination efficiencies than in our study. We find that *Ae. aegypti* populations from the Asia-Pacific region are highly competent to transmit a YFV of the West-African genotype, giving legitimacy to the evaluation of the risk of YF epidemics in this YF-free region. Originally from tropical rainforests in Africa where it circulates between non-human primates and zoophilic mosquitoes, YFV was introduced into the Americas during the slave trade from the 14th century, as was the YFV vector, *Ae. aegypti*[2]. The eradication of *Ae. aegypti* led to the success in controlling YF, but the relaxation of vector control in the 1970s permitted *Ae. aegypti* to recolonize the region[29]. This species then became responsible for urban dengue outbreaks[30] but was excluded from the YFV cycle, mainly sylvatic in South America[31]. Thus, YFV is absent elsewhere in the world except in Africa and in America, until 2016 when 11 YFV-infected workers returning from Angola were reported in China, putting the YF risk back on the agenda[3]. In Asia, all the ingredients to fuel a sylvatic cycle are gathered as well as an urban cycle: 49 of the 52 countries are considered to be suitable for the proliferation of *Ae. aegypti* and/or *Ae. albopictus*[32], offering the fertile ground for YF transmission in addition to dengue fever[33,34]; even though the YFV-susceptible non-human primates of South America are absent in Asia[35], *Macaca* spp. monkeys widely distributed in Asia might have the

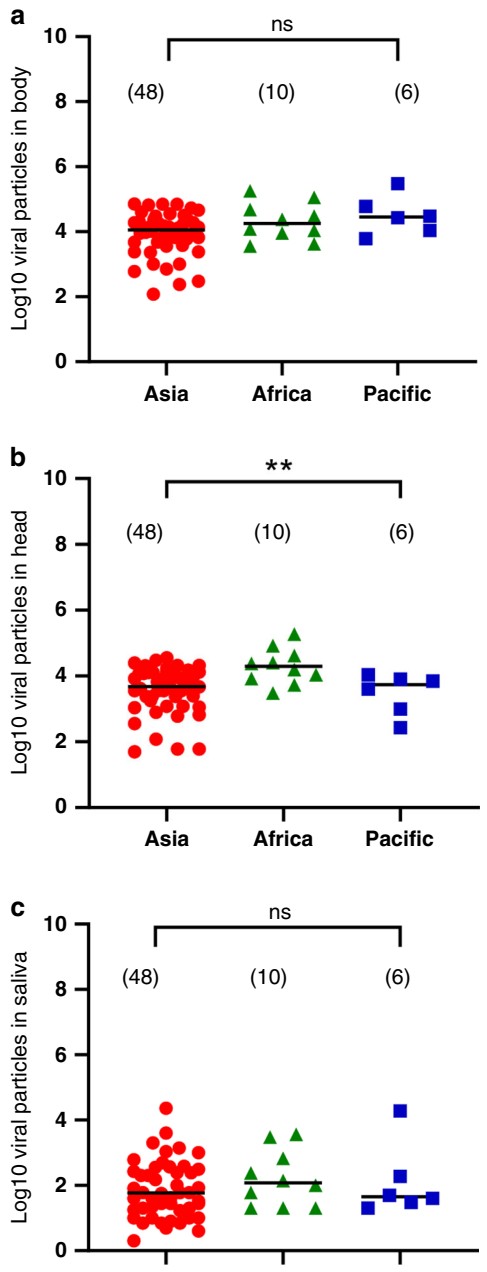

**Fig. 4 Virals loads in body, head, and saliva of *Aedes aegypti* populations from Asia, Africa, and the Pacific region.** Mosquito (**a**) body, (**b**) head, and (**c**) saliva were titrated on C6/36 cells and the number of viral particles was expressed in ffu/sample. Stars indicate statistical significance of comparisons by the Kruskal–Wallis test (two-sided test; **P ≤ 0.01). **b** **P = 0.0095. ns (non-significant) indicates a lack of statistical significance (P > 0.05). Bars indicate the mean. In brackets are the numbers of mosquitoes tested. Red dots: samples from Asia; green triangles: samples from Africa; blue squares: samples from the Pacific region. Source data are provided in Supplementary Data 2 file.

**Table 1 Univariate linear regression analyses for the body, head, and saliva in *Aedes aegypti* mosquitoes, 21 days after the infectious blood meal at a titer of $10^7$ ffu/mL.**

| Continent | Crude coefficient (95% CI) | P Kruskal–Wallis test |
|---|---|---|
| *Body* | | |
| Africa | 1 | 0.11 |
| Asia | −0.39 (−0.83; 0.06) | |
| Pacific | +0.19 (−0.48; 0.85) | |
| *Head* | | |
| Africa | 1 | **0.01** |
| Asia | −0.73 (−1.21; −0.26) | |
| Pacific | −0.82 (−1.53; −0.12) | |
| *Saliva* | | |
| Africa | 1 | 0.54 |
| Asia | −0.35 (−0.97; 0.27) | |
| Pacific | −0.10 (−1.02; 0.82) | |

Analyses were performed according to the continent where mosquitoes were collected. Source data are provided in Supplementary Data 2 file.
In bold, significant values (P ≤ 0.05).

However, the number of viral particles excreted by these mosquitoes is similar to the viral loads estimated from African mosquitoes suggesting that once able to transmit, the Asian-Pacific *Ae. aegypti* mosquitoes are as efficient as mosquitoes from YFV-endemic regions in Africa. To note, we use a West-African YFV isolated in 1979 to infect African mosquitoes from Cameroon and Congo; YFV strains from Senegal show low rates of evolutionary change over time[38] and *Ae. aegypti* from Cameroon, Congo, and Senegal belong to the ancestral form (namely *Ae. aegypti formosus*) and present relatively low levels of genetic differentiation[39] which taken together, limits the bias in estimating vector competence. Laboratory-observed infection experiments show that the proportion of mosquitoes infected and able to transmit YFV was highest for *Ae. aegypti* from Thailand (>50%, Supplementary Fig. 1). Likewise, a recent modeling exercise (data extracted from[28]) suggests that *Ae. aegypti* can commonly be found throughout South-East Asia (Fig. 5a). These results suggest that *Ae. aegypti* from the Asia-Pacific region are competent to YFV and prone to trigger a YF outbreak, strengthening the conclusions drawn from metapopulation models to assess the probabilities of YFV spread based on international airline transportation[40], or disease transmission models using infection data, vaccination coverage, and different environmental factors[41,42]. However, it is important to note that assessing the risk of YFV transmission based on vector competence data as was done in our study conducted in laboratory conditions, does not reflect alone the capacity of mosquitoes to act as a field vector. Some environmental factors might shorten mosquito lifespan and, therefore diminish the probability of infecting after the extrinsic incubation period. Moreover, the viral titers used in our experimental infections may differ from viremias encountered in patients, 4.98 (3.50–5.79) $\log_{10}$ copies/mL of YFV RNA in blood[43]. Finally, viral transmission in our study is determined by detecting viral particles in mosquito saliva collected using the forced salivation technique (see "Methods" section) which do not reflect the physiological dose of viral particles delivered by a mosquito during the bite. Moreover, vector capacity integrates biotic and abiotic factors in addition to vector competence, and therefore, varies in space and time across a region; it can be influenced by population density, vector feeding behavior, and vector lifespan[44]. Apart from making vaccination mandatory, preventing YF outbreaks in the region should rely on controlling *Ae. aegypti* populations, particularly in regions

role of a YFV reservoir[36] alongside YFV-susceptible zoophilic mosquitoes[37].

Interestingly, we show that when infected with a West-African YFV, all *Ae. aegypti* populations examined in this study are able to transmit at day 21 post-infection. With higher rates of dissemination than transmission, our results indicate that the midgut has a less significant role as a physical barrier than the salivary glands.

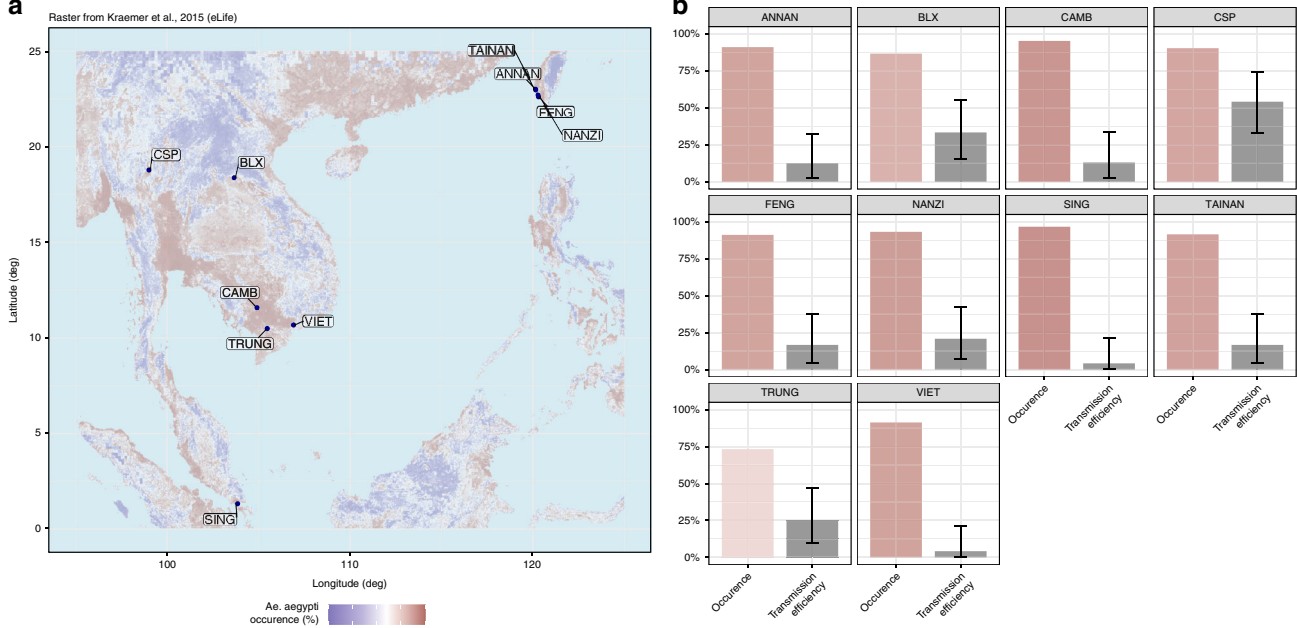

**Fig. 5 Risk of _Aedes aegypti_-mediated YFV transmission in Asia. a** Original data from Kraemer et al. showing the probability of encountering _Aedes aegypti_ in South-East Asia. The colors correspond to probabilities: lower (blue) or higher (red) than the median probability across the whole map (white). **b** Modeled vector occurrence (colored bars matching the values from the scale in **a**) along with mosquito transmission efficiency (gray bar) of _Aedes aegypti_ populations tested in laboratory conditions shown in **a**. This map uses data published by Kraemer et al. and was generated with R v4.0.1 (package raster v3.1-5). Source data are provided in Supplementary Data 1 file[28].

suffering from dengue, chikungunya, and Zika. Although it seems difficult considering the failure in preventing and controlling dengue using conventional insecticides, combining an early detection of imported cases, a vaccination mandatory for travelers returning from countries at risk, a plan for implementing mass vaccination campaigns and securing the vaccine stockpile (still produced in embryonated chicken eggs causing occasional issues of supply), and new promising vector control methods (e.g., _Wolbachia_ strategy) would significantly improve the prevention of YF as of other arboviral diseases[45].

We believe that more work should be done to determine the evolution of viral populations after the escape from the midgut, in the mosquito general cavity where the virus disseminates in various peripheral organs and replicate in disparate tissues. Viral loads in mosquito heads are significantly lower in YFV-infected _Ae. aegypti_ from the Asian-Pacific region suggesting a mechanism able to limit viral replication such as the mosquito immune responses, in particular, the RNA interference, the most important antiviral response against arboviruses[46]. This may refine the mutational spectrum over time, with implications for the diversity of viruses excreted from the mosquito salivary glands and, therefore, viruses injected into the vertebrate host[47]. Other flaviviruses are exclusively endemic to Asia such as the Japanese encephalitis virus (JEV)[48]. It is then legitimate to question if this resident virus might interfere with a non-resident virus, namely YFV[49].

Notably, even if the 17D vaccine has been available since the 1930s, concerns regarding the safety and supply of YFV vaccine make part of the world vulnerable to YF emergence since the manufacturing process of the YF vaccine cannot cover the need for an immediate mass vaccination campaign[13], even though fractional-dose YF vaccination could be an alternative to a shortage of full-dose vaccine[50]. Altogether, our work brings critical data on mosquitoes that deepen our understanding of factors leading to the emergence of arboviruses in order to be better prepared when YF hits the Asia-Pacific region for decision makers[51,52].

## Methods

**Ethics statement.** Animals were housed in the Institut Pasteur animal facilities (Paris) accredited by the French Ministry of Agriculture for performing experiments on live rodents. Work on animals was performed in compliance with French and European regulations on care and protection of laboratory animals (EC Directive 2010/63, French Law 2013-118, February 6th, 2013). All experiments were approved by the Ethics Committee #89 and registered under the reference APAFIS (Autorisation de Projet utilisant des Animaux à des FIns Scientifiques) #6573-2016061412077987 v2.

**YFV strain.** YFV strain S-79 (accession number: MK060080) was isolated from a patient returning from Senegal in 1979, passaged twice on mice brains, and twice on C6/36 cells[53]. Virus stocks for mosquito infections were produced on C6/36 cells and stored at −80 °C until use.

**Mosquito populations.** Twelve _Ae. aegypti_ and six _Ae. albopictus_ populations were analyzed (Table 2 and Fig. 6). Mosquito eggs were collected using ovitraps placed in each locality and shipped to the Institut Pasteur (Paris) for infections. After egg hatching, around 200 larvae were distributed per pan containing one liter of dechlorinated water and yeast tablets as food. Larvae were reared until the adult stage in controlled conditions[54]. OF1 mice for feeding mosquitoes were between 6-week and 2-month-old, maintained in an animal facility under standard conditions (23 °C and 14:10 light/dark cycle) at Institut Pasteur.

**Mosquito infectious blood meal.** Boxes of sixty 10-day-old female adults were transferred into biosafety level-3 (BSL-3) laboratory 24 h prior to infection. The blood meal was composed of 1.4 mL of rabbit erythrocytes supplemented with 10 mM adenosine triphosphate (ATP) as a phagostimulant, and 0.7 mL of viral stock to obtain a final titer of $10^7$ ffu/mL. The infectious blood meal was placed in capsules of a Hemotek® blood-feeding system (Hemotek Ltd, Blackburn, UK) at 37 °C. The engorged mosquitoes were then kept at 28 °C in 80% humidity and fed with a 10% sucrose solution until processing at 14 and 21 days post-infection (dpi). The rabbits used for preparing infectious blood meals were between 3-month and 2-year-old and maintained in an animal facility under standard conditions (23 °C and 14:10 light/dark cycle) at Institut Pasteur.

**Preparation of mosquito samples.** Saliva was collected after removing the wings and legs of mosquitoes and inserting the proboscis into a p20 tip filled with 5 μL of FBS (Fetal Bovine Serum)[55]. After 30 min, the saliva-containing FBS was expelled into 45 μL of L-15 medium and stored at −80 °C until analysis. To determine infection rate (IR) and dissemination rate (DR), bodies and heads were homogenized in 300 μL of L-15 medium supplemented with 2% of FBS. After

**Table 2 Mosquito populations, countries, localities, and generation used.**

| Mosquito species | Country | Population name | Locality | Generation[a] | Date of collection | Collaborations |
|---|---|---|---|---|---|---|
| Aedes aegypti | Cambodia | CAMB | Phnom Penh | 3 | 08.2018 | Boyer S. (Institut Pasteur of Cambodia) |
| | Vietnam | VIET | An Giang | 1 | 08.2018 | Huynh T. (Institut Pasteur of Ho Chi Minh City, Vietnam) |
| | | TRUNG | Trung Muoi | 3 | 10.2018 | |
| | Laos | BLX | Bolikhamxay | 3 | 09.2019 | Marcombe S. (Institut Pasteur of Laos) |
| | Thailand | CSP | Chiang Mai | 4 | 02.2019 | Jupatanakul N. (National Center for Genetic Engineering and Biotechnology, Thailand) |
| | Singapore | SING | Singapore | 1 | 2019 | Pompon J. (National University of Singapore) |
| | Taiwan | FENG | Kaohsiung | 1 | 04.2019 | Chen C.H. (National Health Research Institute, Taiwan) |
| | | NANZI | Kaohsiung | 1 | 04.2019 | |
| | | ANNAN | Tainan | 1 | 04.2019 | |
| | | TAINAN | Tainan | 1 | 04.2019 | |
| | New Caledonia | NOUMEA | Nouméa (quartier Normandie) | 2 | 2019 | Pocquet N. (Institut Pasteur of New Caledonia) |
| | French Polynesia | PAEA | Tahiti | Lab colony | 1994 | Failloux A.B. |
| Aedes albopictus | China | FOSHAN | Guangdong Province | Lab colony | 1981 | Chen X.G. (Southern Medical University, Guangzhou, China) |
| | Japan | YYG | Tokyo | Lab colony | 2014 | Sawabe K. (NIID) |
| | Taiwan | LINGYA | Lingya | 6 | 04.2019 | Chen C.H. (National Health Research Institute, Taiwan) |
| | Laos | XKH | Xieng Khouang | 3 | 09.2019 | Marcombe S. (Institut Pasteur of Laos) |
| | Thailand | THAI | Chiang Mai | 7 | 02.2019 | Jupatanakul N. (National Center for Genetic Engineering and Biotechnology, Thailand) |
| | Brazil | PMNI | Nova Iguaçu | 8 | 2015 | Lourenço-de-Oliveira R. (Instituto Oswaldo Cruz, Brazil) |

aGeneration refers to the generation of mosquitoes after field collection. Lab colony refers to a mosquito strain that has been adapted to laboratory conditions for more than 20 generations.

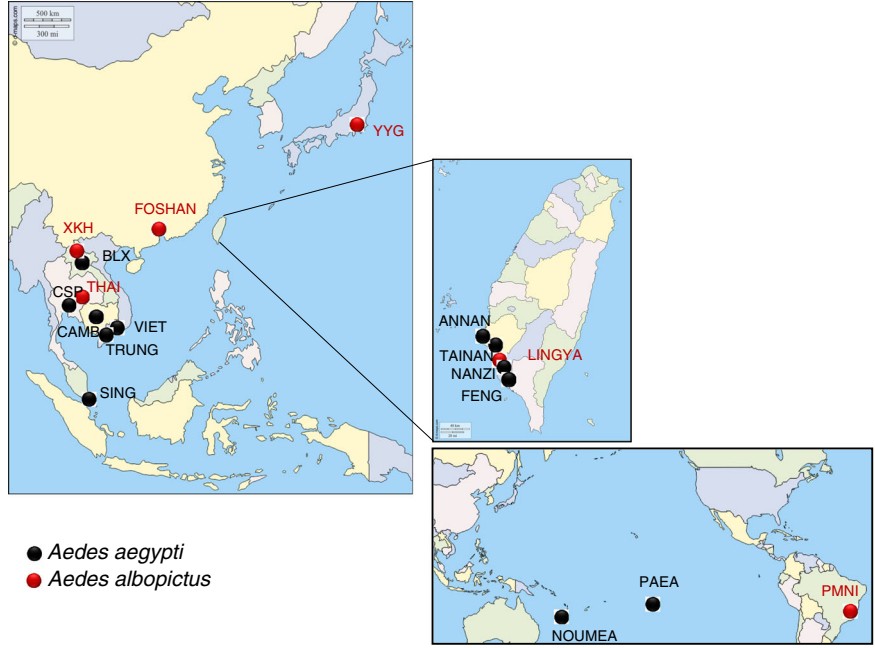

**Fig. 6 Geographical distribution of the 18 mosquito sample locations (12 *Aedes aegypti* and 6 *Aedes albopictus*).** Black dots: *Aedes aegypti*; red dots: *Aedes albopictus*. The map was built using the open-source map site "https://d-maps.com/conditions.php?lang=en/". Each dot corresponds to a sampling location.

centrifugation at 10,000 rpm for 10 min, supernatants were collected for virus detection. Moreover, to study if patterns of infection, dissemination, and transmission were different in *Ae. aegypti* populations from the Asia-Pacific region compared to mosquitoes from YFV-endemic regions in Africa, we included mosquito populations from Cameroon (Benoué, Douala, and Yaoundé) and Congo (Brazzaville) to our dataset; African *Ae. aegypti* analyzed were partly processed in the previous publication of Kamgang et al.[54].

**Virus titration**. Serially diluted samples were inoculated on C6/36 cells in 96-well plates; each well was inoculated with 50 µL of diluted samples for one hour at 28 °C and after removing the inoculum, cells were covered with 150 µL of carboxymethylcellulose (CMC) supplemented with L-15 medium. After incubation at 28 °C for 5 days, cells were fixed with 3.6% formaldehyde, washed and hybridized with YFV specific primary antibody (catalog number: NB100-64510, Novusbio, CO, USA), and revealed by using a fluorescent-conjugated secondary antibody (catalog number: A-11029, Life Technologies, CA, USA), with dilution factors 1:200 and 1:1000, respectively. Foci were counted under a fluorescent microscope and titers were expressed as ffu/sample.

**Risk of *Ae. aegypti*-mediated YFV transmission**. The work by Kraemer et al[28]. presents worldwide estimates of the occurrence of *Ae. aegypti*, i.e., the probability of encountering *Ae. aegypti* throughout at a resolution of 5 km × 5 km. We extracted these values at the sampling points where studied mosquito populations can be found as well as averaged these at each geographical point to illustrate possible heterogeneity in mosquito occurrence.

**Statistical analyses**. IR, DR, and TR were compared among populations using Fisher's exact test. Virus titrations were compared among populations using Kruskal–Wallis non-parametric tests. Correlations between titration in bodies, heads, and saliva were estimated. Statistical analyses were performed using the Stata software (StataCorp LP, Texas, USA). *P*-values < 0.05 were considered statistically significant. If necessary, the significance level of each test was adjusted based on the number of tests run, according to the sequential method of Bonferroni[56]. The statistical details can be found in the figure legends and the effect of geographic origin was estimated using a linear regression model.

**Reporting summary**. Further information on research design is available in the Nature Research Reporting Summary linked to this article.

## Data availability
The original vector distribution maps from Kraemer et al.[28], are provided by their authors online (http://goo.gl/Zl2P7J). The data that support the findings of this study are available as supplementary information files (Supplementary Data 1 and 2).

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

## Acknowledgements
We thank our collaborators cited in Table 2 for mosquito collections. We thank Adrien Blisnick, Rachel Bellone, and Gaelle Gabiane for editing. We also thank Peter Sahlins for correcting english.

## Author contributions
L.G.L. performed the research. M.V. and L.M. provided technical help. T.O. and Y.M. did data modeling and statistical analyses. B.K. participated in interpreting results for mosquitoes from Africa. C.H.C. participated in the research design and funding. P.S.Y. and A.B.F. conceptualized the study, coordinated the research, and wrote the paper. All authors have read and agreed with the published version of the manuscript.

## Competing interests
The authors declare no competing interests.
