## [Peer Review File · Nature Communications]

REVIEWER COMMENTS

Reviewer #1 (Remarks to the Author):

This manuscript by Lataillade and colleagues presents data from vector competence studies from various mosquito population across Asia to assess one aspect of Yellow Fever risk in the region. While the core vector competence data is of interest, the data analysis has some serious flaws and the conclusions of the paper are overstated and not sufficiently supported by the data presented. Large parts of this manuscript need to be modified.

Major comments:

It is not clear how this work fits in with previous vector competence studies on *Ae. aegypti* and *Ae. albopictus* in Asia and there is no reference to previous work in the Introduction or the Discussion. While I believe the authors that this is likely the largest study to date, it is not clear what exactly this study adds over previous work showing that some Asian *Ae. aegypti* populations are competent for YFV

The Asia-wide spatial model has some serious flaws and should be omitted from the analysis entirely. Among them:

- Too few datapoints to reliably extrapolate to continent-level models
- Oversimplistic methods used for extrapolation (inverse distance weighting) that don't take into account any spatial autocorrelation in the data or test for potentially associated covariates.
- The Kraemer et al. *Aedes* maps map the probability of detecting a mosquito population which has a non-linear (and unvalidated) relationship with vector abundance and is unsuitable to treat as such.
- The formula used for vectorial capacity omits several important components and is unreferenced. It makes no allowances for biting behaviour and assumes that all mosquitoes can survive the extrinsic incubation period (would likely to be rare at 14 days and even more rare 21 days)
- No effort is made to appropriately propagate uncertainty in vector competence measures, extrapolation uncertainty, uncertainty from the Kraemer maps or uncertainty from the other relevant parameters in the vector competence equation.

The discussion does not include a limitations section despite the study having many limitations.

Data availability- currently available on "reasonable request" – is there any strong reason why the data cannot be made available from this paper? It would certainly improve reuse. Similar comment with regards to code availability, this is not difficult to share and improves transparency and interpretability of the study.

I believe the conclusions of the paper are substantially overstated. In particular statements such as: "the risk of transmission to human populations is alarmingly high" are not supported by data presented in this paper. Figure 6 suggests nowhere in Asia is risk as high as Africa and at its highest is 50% African levels (outside the Noumea results in the Pacific). 50% reduction in vector competence may be enough for $R_0 < 1$ (would need to be tested in other work). Beyond this, there are a wide range of other reasons why vector competence would be necessary but not sufficient to allow YFV transmission in Asia including but not limited to: levels of cross immunity from other flaviviruses, differing biting behaviours, lack of specific connectivity (Chinese imported cases were to Beijing where *Ae. aegypti* is absent), the unknown role of sylvatic cycles in supporting urban transmission cycles, etc, etc. The fact remains that hundreds of years of epidemiological evidence still goes against the risk of "alarmingly high" transmission in Asia and I don't think this paper presents enough evidence to overturn this.

Minor comments:

Line 126-133- multiple references in the text to wide uncertainty but no figure presented on the range of transmission probabilities predicted.

Statistical tests for geographical differences- there are a lot of hypothesis tests 12 populations * 2 timepoints * 3 measures * 2 species in the first two figures alone. I would question how much weight to put on any significant results given the probability of a significant difference by chance.

Why is no data on IR, DR and TR presented for the African mosquito species?

Figure 6 missing legend- how do you define "able to transmit YFV"

Comment on the likely impact of using YFV from only one old location (Senegal 1979) – positive control (African) mosquito populations are from a very different setting (central vs West Africa) and decades apart

Reviewer #2 (Remarks to the Author):

With increasing global travel, the risk of exportation of yellow fever (YF) out of Africa and Latin America to Asia is increasing. The question therefore remains whether the vectors in Asia are as competent for virus transmission as in Africa and the Americas.

Yellow fever carries a high case fatality rate (CFR); for severe yellow fever cases hospitalized in Brazil during the most recent outbreak, the CFR was above 60% (please quote: Managing severe yellow fever in the intensive care: lessons learnt from Brazil.

Kallas EG, Wilder-Smith A. *J Travel Med.* 2019 Jun 11;26(5):taz043. doi: 10.1093/jtm/taz043.

Severe yellow fever in Brazil: clinical characteristics and management.

Ho YL, Joelsons D, Leite GFC, Malbouisson LMS, Song ATW, Perondi B, Andrade LC, Pinto LF, D'Albuquerque LAC, Segurado AAC; Hospital das Clínicas Yellow Fever Assistance Group. *J Travel Med.* 2019 Jun 11;26(5):taz040. doi: 10.1093/jtm/taz040. PMID: 31150098)

The combination of high CFR and increasing travel necessitates a risk assessment for YF introduction AND establishment in Asia. This paper is clearly an advance to the existing literature on the risk based on vector competence. The authors investigated *Aedes* populations from different Asian countries/cities, and found that the vector competence is basically similar. This is an important message to the international community.

The authors would do well by adding some references on the increasing global travel and vulnerabilities of receiving countries: Global travel patterns: an overview.

Glaesser D, Kester J, Paulose H, Alizadeh A, Valentin B. *J Travel Med.* 2017 Jul 1;24(4). doi: 10.1093/jtm/tax007.

Global trends in air travel: implications for connectivity and resilience to infectious disease threats. Tuite AR, Bhatia D, Moineddin R, Bogoch II, Watts AG, Khan K. *J Travel Med.* 2020 May 6:taaa070. doi: 10.1093/jtm/taaa070.

The authors also compared *Aedes aegypti* versus *albopictus* vector competence and found a striking difference in that *Aedes aegypti* is clearly more competent. As *Aedes aegypti* have now

invaded basically all major cities in Asia---in fact they thrive in urbanized environments (see; Epidemic arboviral diseases: priorities for research and public health. Wilder-Smith A, Gubler DJ, Weaver SC, Monath TP, Heymann DL, Scott TW. Lancet Infect Dis. 2017 Mar;17(3):e101-e106. doi: 10.1016/S1473-3099(16)30518-7.), the risk of importation through travel hub networks and airports is increasing.

Much can be learned from dengue and its rapid geographic expansion around the world. Dengue and YF share the same urban-adapted vector *Aedes aegypti*: Estimating the size of *Aedes aegypti* populations from dengue incidence data: Implications for the risk of yellow fever outbreaks. Massad E, Amaku M, Coutinho FAB, Struchiner CJ, Lopez LF, Wilder-Smith A, Burattini MN. Infect Dis Model. 2017 Dec 8;2(4):441-454. doi: 10.1016/j.idm.2017.12.001. eCollection 2017 Nov. PMID: 30137722

Although YF remains a disease mainly transmitted in sylvatic cycles (unlike dengue that now has become a predominant human-vector-human transmitted disease in highly urbanized areas), the recent outbreaks in DRC in 2016 and in Brazil in 2017-2018 show the potential for YF to be propagated in urbanized areas.

The findings from this paper should alert the community and prompt preparedness planning, as suggested in a perspective article in the Lancet ID (Yellow fever: is Asia prepared for an epidemic? Wilder-Smith A, Lee V, Gubler DJ. Lancet Infect Dis. 2019 Mar;19(3):241-242.)

The authors should elaborate on why YF has not emerged in Asia despite the fact that the vectors are highly competent.

I would suggest that the authors enhance their conclusions/recommendations in the discussion section by referring to above Lancet ID article. What are the next steps? How can Asia enhance their preparedness? How can vector surveillance be strengthened in Asia to include surveillance for YF? What about access to YF vaccines?

Currently, YF vaccines are not available at all in Asia. Maybe fractional YF vaccines may be a solution in times of limited vaccine supply (please refer to: Fractional-dose yellow fever vaccination: an expert review. Roukens AHE, Visser LG. J Travel Med. 2019 Sep 2;26(6):taz024. doi: 10.1093/jtm/taz024. PMID: 30937437)

Reviewer #3 (Remarks to the Author):

Major comments:

1) Additional background information would strengthen the introduction.

- A brief description of factors involved in YFV transmission on other continents, including vector related factors. YFV transmission cycles are described in the discussion, but perhaps some of this information could be shifted to the introduction?
- It might be helpful to briefly mention existing hypotheses as to why YFV transmission has not been reported in the Asia Pacific to date.
- Information for a general audience on the meaning of vector competence and how it is measured.
- Description of what is already known on the topic, i.e., is there previous work on YFV vector competence using mosquito populations from the Asia Pacific region? If not, please state this.

2) In general, more details and justification of methodological choices, and experiments performed are required in the methods section.

In particular, I'm concerned that the analysis in the section "modeling of geographic dissemination risk" is based on an incorrect interpretation of an output from Kraemer et al 2015.

The quantity estimated in this section likely cannot be considered vector competency. The VC equation shows that this quantity is informed by the output from Kraemer et al. The output surface in Kraemer et al represents the probability of species occurrence or habitat suitability, it does not represent an estimate of vector population size, vector density or probability of encountering the vector, as stated by the authors.

Please describe what geocoded locations i represent compared to 5km x 5km geographical coordinates x .

Please describe what the term w_i in the VC equation represents.

Please explain the meaning of "geodesic".

The authors mention that uncertainty is estimated. Is this reported or plotted somewhere?

Please state what program or programming language was used to build the model and run the analysis, and provide the analysis code.

3) Discussion

Contextualisation of the study results is lacking. The discussion would be substantially strengthened if the results of this study were discussed in the context of other studies of YFV vector competence (potentially from other regions) and other estimates of YFV transmission risk in the Asia Pacific region.

A limitations section is lacking. For example, how well do laboratory studies of vector competence translate to the field?

Minor comments:

Line 44. Issues around vaccine distribution and uptake are also important.

Line 50. Surely both outbound and inbound travel volumes are relevant here?

Line 52. It would strengthen the article to provide more context to these importation events. For example, that they occurred during the 2015-16 YFV outbreak originating in Angola.

Lines 54-58:

Finding i) - in mosquito populations sampled from the Asia-Pacific?

ii) Please state which species.

iii) What does mean from a vector competence/transmission point of view? Perhaps also important to state that this is in post-infection in a laboratory setting.

Lines 63-68. I note that "dpi", "IR" and "DR" are described in the figure captions, but it would be helpful to write these in full the first time they are mentioned in the results section.

Line 81. "All Ae. aegypti mosquitoes". Please consider revising this broad statement, i.e., all Ae

aegypti mosquito populations examined in this study were competent vectors...”

Line 98. Please describe briefly what viral loads are relevant for onward transmission.

Line 128. “Mosquito population sizes (Kraemer et al 2015)”. The outputs from the work presented in this reference do not represent vector population sizes but rather vector habitat suitability. Therefore I’m concerned that the results in this section may not be valid.

Was a formal sample size calculation performed to support any of the statistical analyses?

Table 1. Would be helpful to report the month of collection and sample size. What does generation mean? What does lab colony mean?

Figure 1. Should this be 18 mosquito sampling sites?

Figure 7. As above, Kraemer et al did not estimate vector density.

Answers to Reviewer #1

Point #1: It is not clear how this work fits in with previous vector competence studies on *Ae. aegypti* and *Ae. albopictus* in Asia and there is no reference to previous work in the Introduction or the Discussion. While I believe the authors that this is likely the largest study to date, it is not clear what exactly this study adds over previous work showing that some Asian *Ae. aegypti* populations are competent for YFV

We have added few sentences in the text in **Lines 72-77**: “Although the vector competence for YFV of mosquitoes in Africa, South America, and Caribbean regions, has been investigated^{25,26}, only limited information for Asian-Pacific mosquitoes could be found to measure the possible risk of YFV transmission in this region^{27,28}. Investigating the vector competence for YFV of mosquitoes in the Asia-Pacific region is essential to assess the potential threat of YFV transmission in a region where YF outbreaks have never been reported²⁹.”

- In the reference **25** (*PLoS Negl Trop Dis* 13, e0007783, doi:10.1371/journal.pntd.0007783), Miott et al. (2019) showed that two laboratory colonies of *Ae. malayensis* (more than 50 generations in lab) from Singapore and *Ae. aegypti* (generation 8) from Laos infected with a YFV (West African genotype) were able to transmit YFV by detecting viral particles in mosquito saliva.
- In the reference **26** (*Mem Inst Oswaldo Cruz* 97, 437-439), Lourenco-de-Oliveira et al. (2002) examined *Ae. aegypti* from Cambodia (1), Vietnam (2) and French Polynesia (1) and showed that these mosquitoes were able to disseminate YFV.

Point #2: The Asia-wide spatial model has some serious flaws and should be omitted from the analysis entirely. Among them:

- **Too few datapoints to reliably extrapolate to continent-level models**
- **Oversimplistic methods used for extrapolation (inverse distance weighting) that don't take into account any spatial autocorrelation in the data or test for potentially associated covariates.**
- **The Kraemer et al. Aedes maps map the probability of detecting a mosquito population which has a non-linear (and unvalidated) relationship with vector abundance and is unsuitable to treat as such.**
- **The formula used for vectorial capacity omits several important components and is unreferenced. It makes no allowances for biting behaviour and assumes that all mosquitoes can survive the extrinsic incubation period (would likely to be rare at 14 days and even more rare 21 days)**
- **No effort is made to appropriately propagate uncertainty in vector competence measures, extrapolation uncertainty, uncertainty from the Kraemer maps or uncertainty from the other relevant parameters in the vector competence equation.**

We thank the reviewer for raising concerns and highlighting that this section should be improved. It is true than inverse distance weighting models are simplistic by design, as they estimate a smoothed surface by interpolating between a set of observed data points. We have made several changes to the text to improve the description of the model parameterization and the nature of the estimated raster. We have added in **Lines 267-282** to clarify notations of geocoding and now we state that Kraemer et al. maps present “the probability of encountering mosquitoes”. We also detailed how model parameterization was chosen to limit the influence of observed values for a more conservative approach.

Furthermore, considering the moderate number of sampled mosquito populations, we decided to restrict this analysis to the Asia-Pacific region only. In particular, we also excluded Japan, where only an *Ae. albopictus* lab colony was studied and increased spatial spread of data unreasonably. As such,

we refrained from extrapolating to continent-wide estimates but rather restricted that analysis to a subset of the Asian continent.

We however disagree that such models are not suited for the presented data, due to (1) possible spatial autocorrelation or (2) potentially associated covariates. With respect to **(1)**, inverse distance weighting is actually built on the postulate that data points are spatially correlated: the closer observed points are, the more they will contribute to estimated values around them. The underlying assumption in our work is that *Aedes* mosquitoes at a given location are more similar in behavior and disease susceptibility to those from another nearby location, than to others found hundreds of kilometers away. This is supported by entomological observations for urban and peri-urban *Aedes* spp., e.g. O'Neill et al. 2020 (<https://doi.org/10.1371/journal.pntd.0008157>). The concern for geographic areas with vectors competent for a yet-unintroduced disease has been often raised over the past decade (see Eisenberg et al. 2013 (<https://doi.org/10.4269/ajtmh.12-0485>), O'Neill et al. 2017 (<https://doi.org/10.1371/journal.pntd.0005496>); Higgs et al. 2015 (<https://doi.org/10.1089/vbz.2015.9001.int>); Hayes et al. 2009 (<https://doi.org/10.1056/NEJMoa0805715>)). Regarding **(2)** and contrary to regression techniques, inverse distance weighting did not involve adjusting on explanatory covariates but only assumes contribution from geographical coordinates.

As pointed out by the reviewer, the maps published by Kraemer et al. provide values between 0 and 1 that their authors describe as follows: “The map depicts the probability of occurrence (from 0 blue to 1 red) at a spatial resolution of 5 km × 5 km”. These can be interpreted as the probability of encountering *Aedes* spp. at any pixel on the raster, each pixel depicting a location of the abovementioned resolution. These probabilities were generated through a planet-wide modeling exercise, with boosted regression trees informed by thousands of sampling points (mosquito occurrence) and multiple covariates (land cover, precipitation, temperature...) with contribution varying geographically. We did not intend to use these probabilities to reflect vector abundance and agree that the relationship between mosquito population size and probability of encountering a mosquito are unknown to this work. However, we used this quantity to reflect human exposure to mosquitoes. We now acknowledge in the main text that our estimates relate to vector competence – an evaluation of the vector capability to transmit the pathogen, rather than vector capacity.

- **Lines 279-281:** “With the present definition, vector competence presents the probability for a mosquito to infect itself upon feeding on human blood and be able to transmit YFV back to a human host.”
- **Lines 501-506:** Legend of Figure 7 was edited to better explain the data provided by Kraemer et al.

In particular, this should address the concerns about mosquito surviving their extrinsic incubation period and living up to D14 or D21, corresponding to dissections conducted in the laboratory to assess infection status.

Finally, we have addressed the reviewer’s concern about estimating uncertainty in our models by providing the estimated maps of uncertainty in Fig. S1. We also detailed in the main text how these maps were generated:

- **Lines 284-286:** “Uncertainty for the observed proportion of mosquitoes able to disseminate YFV was defined as the width of its 95% binomial confidence interval and the same spatial model was reproduced with this measure as outcome to estimate spatial uncertainty.”
- **Line 159:** Added reference to Fig. S5

Point #3: The discussion does not include a limitations section despite the study having many limitations.

We have added the following statement in **Lines 200-204**: “However, it is important to note that the risk of transmission model based on vector competence data does not reflect alone the capacity of mosquitoes to act as a field vector. Thus vector capacity integrates biotic and abiotic factors in addition to vector competence, and therefore, varies in space and time across a region; it can be influenced by population density, vector feeding behavior, and vector lifespan⁴².”

Point #4: Data availability- currently available on “reasonable request” – is there any strong reason why the data cannot be made available from this paper? It would certainly improve reuse. Similar comment with regards to code availability, this is not difficult to share and improves transparency and interpretability of the study.

We acknowledge the strong urge for open science and try and commit to FAIR principles as much as possible. We now provide the full code used to conduct the spatial analyses as part of Electronic Supplementary Material ESM1. This supplementary material consists of:

- a .RData file enclosing the curated results of the laboratory experiments. Curation only consisted in recoding/renaming raw data and no data points used in the analysis were removed. Only unused points from day 14 post-infection were removed from that table;
- A comprehensive .R file providing reusable code for the spatial analysis

We have edited the text to reflect availability of these electronic resource files:

- **Lines 296-297**: “The data that support the findings of this study are available in the Electronic Supplementary Material ESM1. The data can be loaded into R from the attached .RData binary file.”
- **Lines 299-300**: “All analyses conducted in this manuscript and associated code will be made available upon request to the authors. The code developed for conducting the spatial analysis is available in ESM1 as a comprehensive .R file.”

Point #5: I believe the conclusions of the paper are substantially overstated. In particular statements such as: “the risk of transmission to human populations is alarmingly high “ are not supported by data presented in this paper. Figure 6 suggests nowhere in Asia is risk as high as Africa and at its highest is 50% African levels (outside the Noumea results in the Pacific). 50% reduction in vector competence may be enough for $R_0 < 1$ (would need to be tested in other work). Beyond this, there are a wide range of other reasons why vector competence would be necessary but not sufficient to allow YFV transmission in Asia including but not limited to: levels of cross immunity from other flaviviruses, differing biting behaviours, lack of specific connectivity (Chinese imported cases were to Beijing where *Ae. aegypti* is absent), the unknown role of sylvatic cycles in supporting urban transmission cycles, etc, etc. The fact remains that hundreds of years of epidemiological evidence still goes against the risk of “alarmingly high” transmission in Asia and I don’t think this paper presents enough evidence to overturn this.

The word “alarmingly” has been removed. We have also toned down the interpretation of some of our findings regarding *Aedes* vector competence in south-east Asia. Although the mosquito vector competence cannot be used alone to estimate the disease risk on the field, the strength of this study lies in testing the YFV susceptibility of multiple mosquito populations in the Asia-Pacific region in the same conditions (virus titer, dose and procedures), making comparisons more robust.

Point #6: Line 126-133- multiple references in the text to wide uncertainty but no figure presented on the range of transmission probabilities predicted.

We thank the reviewer for pointing out that uncertainty map was referenced but not presented. This is now addressed as Fig. S5. We have modified the text accordingly, as mentioned in response to Point #2.

Point #7: Statistical tests for geographical differences- there are a lot of hypothesis tests 12 populations * 2 timepoints * 3 measures * 2 species in the first two figures alone. I would question how much weight to put on any significant results given the probability of a significant difference by chance.

When a set of comparisons are run simultaneously, the Bonferroni correction has been used; it lowers the α value taking into account the number of comparisons performed.

We have added in **Lines 291-292**: “If necessary, the significance level of each test was adjusted based on the number of tests run, according to the sequential method of Bonferroni⁵⁵”

Point #8: Why is no data on IR, DR and TR presented for the African mosquito species?

Data on IR, DR and TR for mosquitoes from Africa are available in the paper of Kamgang et al. 2019. Potential of *Aedes albopictus* and *Aedes aegypti* (Diptera: Culicidae) to transmit yellow fever virus in urban areas in Central Africa. *Emerging Microbes & Infections* 8:1, 1636-1641, DOI:10.1080/22221751.2019.1688097.

From this study, we used the mosquitoes presenting infectious saliva and titrated body and head. It has been stated in the text in **Lines 254-258**: “Moreover, to study if patterns of infection, dissemination and transmission were different in *Ae. aegypti* populations from the Asia-Pacific region compared to mosquitoes from YFV-endemic regions in Africa, we included mosquito populations from Cameroon (Benoué, Douala, and Yaoundé) and Congo (Brazzaville) to our dataset; African *Ae. aegypti* analyzed were partly processed in the previous publication of Kamgang et al.⁵².”

Point #9: Figure 6 missing legend- how do you define “able to transmit YFV”

We have added this definition as part of the legend, and also rephrased the main legend to avoid confusion between “ability to transmit” –which may involve the notion of inoculum, and technical ability to detect the virus with laboratory-based techniques. We wrote: “Proportion of *Aedes aegypti* mosquitoes with positive YFV titers in saliva. Ability to transmit YFV was defined as the proportion of mosquitoes that had detectable viruses in saliva after dissection in the laboratory.”

Point #10: Comment on the likely impact of using YFV from only one old location (Senegal 1979) – positive control (African) mosquito populations are from a very different setting (central vs West Africa) and decades apart.

We have added these sentences in **Lines 192-196**: “To note, we used a West African YFV isolated in 1979 to infect African mosquitoes from Cameroon and Congo; YFV strains from Senegal show low rates of evolutionary change over time⁴⁰, and *Ae. aegypti* from Cameroon, Congo, and Senegal belong to the ancestral form (namely *Ae. aegypti formosus*) and present relatively low level of genetic differentiation⁴¹ which taken together, limits the bias in estimating vector competence.”

Answers to Reviewer #2

Point #1: Yellow fever carries a high case fatality rate (CFR); for severe yellow fever cases hospitalized in Brazil during the most recent outbreak, the CFR was above 60% (please quote: Managing severe yellow fever in the intensive care: lessons learnt from Brazil. Kallas EG, Wilder-Smith A. J Travel Med. 2019 Jun 11;26(5):taz043. doi: 10.1093/jtm/taz043. Severe yellow fever in Brazil: clinical characteristics and management. Ho YL, Joelsons D, Leite GFC, Malbouisson LMS, Song ATW, Perondi B, Andrade LC, Pinto LF, D’Albuquerque LAC, Segurado AAC; Hospital das

Clínicas Yellow Fever Assistance Group. J Travel Med. 2019 Jun 11;26(5):taz040. doi: 10.1093/jtm/taz040. PMID: 31150098)

We have added in **Lines 39-41**: “Similar to other flaviviruses, the common symptoms of YF are fever, headache, muscle aches, nausea, and vomiting, however, the in-hospital case fatality rate (CFR) could reach 67%^{7,8}, giving this disease a particular interest for public health.”

Point #2: The authors would do well by adding some references on the increasing global travel and vulnerabilities of receiving countries:

Global travel patterns: an overview. Glaesser D, Kester J, Paulose H, Alizadeh A, Valentin B. J Travel Med. 2017 Jul 1;24(4). doi: 10.1093/jtm/tax007.

Global trends in air travel: implications for connectivity and resilience to infectious disease threats. Tuite AR, Bhatia D, Moineddin R, Bogoch II, Watts AG, Khan K. J Travel Med. 2020 May 6:taaa070. doi: 10.1093/jtm/taaa070.

The two references have been added in **Lines 64-66**: “Notable increase of travels between countries with different capacities to detect and control infectious diseases (e.g. growth of tourism in emerging countries) can facilitate the geographic spread of vector-borne diseases ^{21,22}.”

Point #3: Much can be learned from dengue and its rapid geographic expansion around the world. Dengue and YF share the same urban-adapted vector Aedes aegypti: Estimating the size of Aedes aegypti populations from dengue incidence data: Implications for the risk of yellow fever outbreaks. Massad E, Amaku M, Coutinho FAB, Struchiner CJ, Lopez LF, Wilder-Smith A, Burattini MN. Infect Dis Model. 2017 Dec 8;2(4):441-454. doi: 10.1016/j.idm.2017.12.001. eCollection 2017 Nov. PMID: 30137722

The reference Massad et al (2017) has been cited in **Line 183** (reference 36).

Point #4: The findings from this paper should alert the community and prompt preparedness planning, as suggested in a perspective article in the Lancet ID (Yellow fever: is Asia prepared for an epidemic? Wilder-Smith A, Lee V, Gubler DJ. Lancet Infect Dis. 2019 Mar;19(3):241-242.)

The reference Wilder-Smith et al (2019) has been cited in **Line 77** (reference 29).

Point #5: The authors should elaborate on why YF has not emerged in Asia despite the fact that the vectors are highly competent.

Assessing risks of transmission and emergence is usually done by evaluating the vector competence. However, it remains an experimental measure that should be included in a formula named the vectorial capacity. The vectorial capacity is influenced by vector densities, blood feeding behavior and vector longevity. For example, a highly competent mosquito can be an inefficient vector if it is not anthropophilic (does not feed on humans) or if it has a limited lifespan, too short to become infectious and transmit. Conversely, a poorly competent vector can cause an outbreak if the infected vector can become infectious very quickly or is highly anthropophilic and present in high densities in domestic environments.

These statements have been integrated in the ms in **Lines 200-204**: “However, it is important to note that the risk of transmission model based on vector competence data does not reflect alone the capacity of mosquitoes to act as a field vector. Thus vector capacity integrates biotic and abiotic factors in addition to vector competence, and therefore, varies in space and time across a region; it can be influenced by population density, vector feeding behavior, and vector lifespan⁴².”

Point #6: I would suggest that the authors enhance their conclusions/recommendations in the discussion section by referring to above Lancet ID article. What are the next steps? How can Asia enhance their preparedness? How can vector surveillance be strengthened in Asia to include surveillance for YF? What about access to YF vaccines? Currently, YF vaccines are not available at all in Asia. Maybe fractional YF vaccines may be a solution in times of limited vaccine supply (please refer to: Fractional-dose yellow fever vaccination: an expert review. Roukens AHE, Visser LG. J Travel Med. 2019 Sep 2;26(6):taz024. doi: 10.1093/jtm/taz024. PMID: 30937437)

We have considered the reviewer's suggestions and have added to:

- **Lines 204-209:** "Apart from making vaccination mandatory, preventing YF outbreaks in the region should rely on controlling *Ae. aegypti* populations, which seems difficult considering the failure in preventing and controlling dengue using conventional insecticides. Therefore, combining a secured production of YFV vaccine (still produced in embryonated chicken eggs causing occasional issues of supply) and new promising vector control methods (e.g. *Wolbachia* strategy) would significantly improve the prevention and control of YF as of dengue⁴³."
- **Lines 221-222:** "even though fractional-dose YF vaccination could be an alternative to a shortage of full-dose vaccine⁴⁸."

Answers to Reviewer #3

Point #1: 1) Additional background information would strengthen the introduction.

- A brief description of factors involved in YFV transmission on other continents, including vector related factors. YFV transmission cycles are described in the discussion, but perhaps some of this information could be shifted to the introduction?

- It might be helpful to briefly mention existing hypotheses as to why YFV transmission has not been reported in the Asia Pacific to date.

- Information for a general audience on the meaning of vector competence and how it is measured.

- Description of what is already known on the topic, i.e., is there previous work on YFV vector competence using mosquito populations from the Asia Pacific region? If not, please state this.

As suggested, we have added more background information in the introduction:

- **Lines 35-38:** "(e.g. transmission barrier resulting from a low compatibility between mosquito and virus genotypes^{2,3}, limited duration and low viraemia in humans, absence of a sylvatic cycle^{4,5}, competition with well-established flaviviruses as dengue and Japanese encephalitis viruses⁶) are still poorly explored,"
- **Lines 50-58:** "To transmit an arbovirus such as YFV, the mosquito should acquire the virus by ingesting a viremic blood from an infected host, the virus enters into the midgut epithelial cells and replicates. After few days of incubation, the virus should pass through the midgut basal lamina and disseminate into the hemocele, then it infects the salivary glands for transmission to the vertebrate host¹⁵. Parameters such as midgut infection, viral dissemination in hemocele, and transmission through infectious saliva are used to determine mosquito vector competence, which is an indicator of transmission risk¹⁶. In Africa and South America, YFV is primarily transmitted in a forest cycle between non-human primates (NHP) and zoophilic mosquitoes (*Aedes* in Africa and *Haemagogus/Sabethes* in South America). Urban cycle of YFV involves mainly the mosquito *Ae. aegypti* in both Africa and South America¹⁷."
- **Lines 72-77:** "Although the vector competence for YFV of mosquitoes in Africa, South America, and Caribbean regions, has been investigated^{25,26}, only limited information for Asian-Pacific mosquitoes could be found to measure the possible risk of YFV transmission in this region^{27,28}. Investigating the vector competence for YFV of mosquitoes in the Asia-Pacific

region is essential to assess the potential threat of YFV transmission in a region where YF outbreaks have never been reported²⁹.”

Point #2: 2) In general, more details and justification of methodological choices, and experiments performed are required in the methods section.

In particular, I'm concerned that the analysis in the section “modeling of geographic dissemination risk” is based on an incorrect interpretation of an output from Kraemer et al 2015.

The quantity estimated in this section likely cannot be considered vector competency. The VC equation shows that this quantity is informed by the output from Kraemer et al. The output surface in Kraemer et al represents the probability of species occurrence or habitat suitability, it does not represent an estimate of vector population size, vector density or probability of encountering the vector, as stated by the authors.

We agree with the reviewer that more consistent phrasing was needed with respect to the data generated by Kraemer et al. and how these were used in the spatial analysis we presented. In particular, we have changed throughout the text improper use of the words “density” and “population size” in place of “probability of encountering” the vector *Aedes spp*.

We want to make it clear that in our approach, we did not misinterpret the maps from Kraemer et al. and always used them for their provided value, that is, “the probability of occurrence [of *Aedes spp*.] (from 0 blue to 1 red) at a spatial resolution of 5 km × 5 km.” as per their author. The quantity therefore estimated by our spatial analysis represents the probability for a mosquito, at a given location, to get infected from human blood and be able to transmit it back to another human upon biting. This is in line with the definition of vector competence as formulated by Kramer 2016 (<https://doi.org/10.1016/j.coviro.2016.08.008>).

Furthermore, we have modified substantially the methods for the spatial modelling to make notations more consistent and for overall readability. (see Points #3–#7).

Point #3: Please describe what geocoded locations *i* represent compared to 5km x 5km geographical coordinates *x*.

The geocoded location *i* and *x* were a simplified references to a set of GPS coordinates consisting of latitude and longitude. Locations denoted with subscript *i* denoted these where mosquito populations were sampled, whereas the generic *x* notation denoted any location on the raster. The text has been modified in **Lines 269-278**: “Let $d(i)$ denote the probability for a mosquito at geocoded location $i = (lat_i, lon_i)$ to disseminate YFV back into humans. An inverse distance weighting spatial model was fitted to the rasterized polygon of sampled locations. The output estimated the complete raster \hat{d} as a smoothed surface of interpolated ability to transmit YFV with a 5 km x 5 km resolution matching that presented by Kraemer et al.³⁰ for the probability of encountering mosquitoes. Vector competence was derived by combining these estimates with the maps $m(x)$ of probabilities of encountering *Aedes spp* from Kraemer et al.³⁰ at any location $x = (lat, lon)$, yielding an overview of mosquitoes' ability to get infected from human blood and transmit back in the general population. The resulting surface can be written as:

$$VC(x) = m(x) * \frac{\sum_i w_i(x) * d(i)}{\sum_i w_i(x)}$$
, with $w_i(x) = \frac{1}{dist(x, i)^p}$, taking interpolated value at every location in the raster except at those observed, remaining unchanged.”

Point #4: Please describe what the term w_i in the VC equation represents.

The term w_i denotes the contribution of known values (at locations *i*) in the process of estimating unknown values (at location *x*). We have added that clarification in the main text in **Lines 278–279**:

“In the above expression $w_i(x)$ denotes the relative contribution of observed value at location i in estimating the value at location x .”

Point #5: Please explain the meaning of “geodesic”.

This is now explained in the text in **Lines 283-284**: “Distances were computed as geodesic to account for earth curvature at such geographical scale.”

Point #6: The authors mention that uncertainty is estimated. Is this reported or plotted somewhere?

We thank the reviewer for pointing out that uncertainty map was not provided. It is now referenced in the Results (**Line 159**) as Fig. S5.

Point #7: Please state what program or programming language was used to build the model and run the analysis, and provide the analysis code.

We agree that code availability is critical and have prepared an Electronic Supplementary Material now referenced in the manuscript as ESM1 and modified the data availability and code availability sections:

- **Lines 296-297**: “The data that support the findings of this study are available in the Electronic Supplementary Material ESM1. The data can be loaded into R from the attached .RData binary file.”
- **Lines 299-300**: “All analyses conducted in this manuscript and associated code will be made available upon request to the authors. The code developed for conducting the spatial analysis is available in ESM1 as a comprehensive .R file.”

The whole spatial analysis can be re-run using the R programming language, provided that one would download the original raster files from Kraemer et al. Direct links to the original data is provided. We also included in that ESM1 the raw vector competence data generated in the laboratory with our experiments.

Point #8: 3) Discussion

Contextualisation of the study results is lacking. The discussion would be substantially strengthened if the results of this study were discussed in the context of other studies of YFV vector competence (potentially from other regions) and other estimates of YFV transmission risk in the Asia Pacific region.

A limitations section is lacking. For example, how well do laboratory studies of vector competence translate to the field?

We have added in **Lines 200-209**: “However, it is important to note that the risk of transmission model based on vector competence data does not reflect alone the capacity of mosquitoes to act as a field vector. Thus vector capacity integrates biotic and abiotic factors in addition to vector competence, and therefore, varies in space and time across a region; it can be influenced by population density, vector feeding behavior, and vector lifespan⁴². Apart from making vaccination mandatory, preventing YF outbreaks in the region should rely on controlling *Ae. aegypti* populations, which seems difficult considering the failure in preventing and controlling dengue using conventional insecticides. Therefore, combining a secured production of YFV vaccine (still produced in embryonated chicken eggs causing occasional issues of supply) and new promising vector control methods (e.g. *Wolbachia* strategy) would significantly improve the prevention and control of YF as of dengue⁴³.”

Point #9: Line 44. Issues around vaccine distribution and uptake are also important.

We have added this information in **Lines 48-49**: “Insecticide-resistance of mosquito populations, as well as a supply shortage, distribution, and uptake of YFV vaccines, are among the main causes of this current burden¹⁴.”

Point #10: Line 50. Surely both outbound and inbound travel volumes are relevant here?

We have removed “outbound” and added the following sentence in **Lines 64-66**: “Notable increase of travels between countries with different capacities to detect and control infectious diseases (e.g. growth of tourism in emerging countries) can facilitate the geographic spread of vector-borne diseases^{21,22}.”

Point #11: Line 52. It would strengthen the article to provide more context to these importation events. For example, that they occurred during the 2015-16 YFV outbreak originating in Angola.

As suggested, we have added more information on the risk of YF in Asia to **Lines 67-72**: “Of greater concern was the report of YFV laboratory-confirmed cases among Chinese travelers returning Asia after a stay in Angola during the 2015-2016 YF outbreak²³, threatening billions of immunologically naïve population in Asia living in close vicinity of *Ae. aegypti* and *Ae. albopictus* mosquitoes¹. Africa receives a large number of Chinese workers who are usually unvaccinated against YFV, increasing the risk of importing YF in Asia²⁴. The combination of repeated introductions of viraemic travelers and immunologically naïve local population in an environment suitable to transmission accentuates the risk of YF emergence in Asia.”

Point #12: Lines 54-58: Finding

- in mosquito populations sampled from the Asia-Pacific?

ii) Please state which species.

iii) What does mean from a vector competence/transmission point of view? Perhaps also important to state that this is in post-infection in a laboratory setting.

We have modified this paragraph (**Lines 78-82**): “We demonstrated that (i) *Ae. aegypti* mosquitoes from the Asia-Pacific region are more susceptible to the West African genotype of YFV than *Ae. albopictus*, (ii) mosquitoes from Singapore, Taiwan, Thailand, and New Caledonia are capable of transmitting YFV at 14 days post-infection, and (iii) *Ae. aegypti* mosquitoes excrete up to 23,000 viral particles in saliva, suggesting that YFV could be transmitted through saliva of infected *Ae. aegypti* in laboratory conditions.”

Point #13: Lines 63-68. I note that “dpi”, “IR” and “DR” are described in the figure captions, but it would be helpful to write these in full the first time they are mentioned in the results section.

It has been done.

Point #14: Line 81. “All *Ae. aegypti* mosquitoes”. Please consider revising this broad statement, i.e., all *Ae. aegypti* mosquito populations examined in this study were competent vectors...”

We have added: “examined in this study” in **Line 105**.

Point #15: Line 98. Please describe briefly what viral loads are relevant for onward transmission.

We have changed the sub title: “Higher loads of viral particles excreted in saliva of *Ae. aegypti* than *Ae. albopictus*”

We need more details to respond properly to the reviewer’s request.

Point #16: Line 128. “Mosquito population sizes (Kraemer et al 2015)”. The outputs from the work presented in this reference do not represent vector population sizes but rather vector habitat suitability. Therefore I’m concerned that the results in this section may not be valid.

We have addressed these remarks as part of our response to Point #2.

Point #17: Was a formal sample size calculation performed to support any of the statistical analyses?

The sample size was defined from our past experience in running vector competence studies with chikungunya and dengue viruses. For yellow fever virus, no sample size was calculated as this was an exploratory work and there were no prior information available that would have suggested point-estimate values of ability to disseminate, and so the point was not to detect a minimum difference but rather describe the current status in the region.

Point #18: Table 1. Would be helpful to report the month of collection and sample size. What does generation mean? What does lab colony mean?

The requested information has been added in the table.

Point #19: Figure 1. Should this be 18 mosquito sampling sites?

Yes, each dot corresponds to one collection site. The legend has been slightly rephrased to make it clear that each dot is a sampling location.

Point #20: Figure 7. As above, Kraemer et al did not estimate vector density.

We have addressed these remarks as part of our response to Point #2.

REVIEWER COMMENTS

Reviewer #1 (Remarks to the Author):

The majority of my points have been addressed and I commend the authors for making fully available the data and code for their analyses and for making greater efforts to display uncertainty. I do, however, still feel that my points about the validity of the geographic extrapolation of results and vector competence calculation (point #2) have been insufficiently addressed.

Point 2.1 too few data points:

Geographic restriction of the analysis has improved this part of the analysis, but poor geographic coverage remains a key issue. There are only 10 geographically distinct points (with 4 very closely located in Taiwan) with predictions made thousands of miles (and in completely different countries) from the data included in this analysis. Any attempt at extrapolation gives, in my view, a false impression of an understanding of the determinants of geographic variation and geographic precision in prediction.

Point 2.2 Methods for extrapolation:

Apologies I probably should have been clearer in my original review. Inverse distance weighting imposes (i.e. assumes) a very strong unfitted functional form on spatial autocorrelation. Ideally this assumption would be supported by an empirical analysis of the spatial autocorrelation of the data e.g. a semi variogram. If this shows that there is no spatial autocorrelation the authors will need to examine regression-based approaches using covariates or remove the extrapolation part of this analysis entirely and just calculate vector competence for the field sites where they do have data.

Point 2.3 Kramer maps

The changes are largely appropriate, although statements like “we used this quantity to reflect human exposure to mosquitoes” are still not inkeeping with interpretations of this map as “human exposure to mosquitoes” is dependent both on presence of a vector population, its abundance and human biting rate.

Point 2.4 Vectorial capacity equation

Despite renaming this measure to “vector competence” I still take issue with the definition of “vector competence” as “an overview of mosquitoes’ ability to get infected from human blood and transmit back in the general population” because such a measure would still need to include the probability of surviving the EIP and the human biting probability. As it currently stands this formula only measures the probability of a single mosquito being infected, not the probability of transmission back to a human population.

Point 2.5 Uncertainty

Good to see the new maps of uncertainty – maybe add 95% CIs when presenting these figures in the relevant results section?

I think more fundamentally, the authors should consider what value this part of the analysis adds to their overall study and if the assumptions they are making here detract and undermine the findings from the rest of their analysis. If “vector competence” (or a modified version of it- point 2.4) were calculated just for the sites where data were available and no extrapolation was attempted would this qualitatively affect the results of this paper? Looking at the text for this section in the results section I don’t think anything would change. I should re-iterate my support for the quality of the rest of the analysis and manuscript, but my view that the geographic extrapolation component of this work is methodologically insufficient, misleading and unnecessary to support the main conclusions of the work remains.

Reviewer #3 (Remarks to the Author):

Thank you for your responses and for the revised manuscript. I have a number of follow-up comments and I still have concerns with the geographic transmission risk model.

In particular, probability of occurrence estimated by Kraemer et al should not be interpreted as the probability of encountering a mosquito. The probability of a human encountering a mosquito would also require information on vector abundance and human population density – and so I am not convinced that vector competence (as per the definition on line 279) has been estimated.

Other specific comments:

1) Please add a description of the findings from reference 27 and compare to the current study results (as done for reference 28).

2) Contextualisation of results from the risk estimates is still lacking from the discussion. There are other published studies which estimate the risk of YF transmission in the Asia Pacific, please discuss your results in the context of these.

3) The new limitations section only discusses limitations related to the transmission risk model. Limitations of the laboratory experiments and analyses should also be discussed.

4) Lines 204-206. "Apart from making vaccination mandatory, preventing YF outbreaks in the region should rely on controlling *Ae. aegypti* populations, which seems difficult considering the failure in preventing and controlling dengue using conventional insecticides.

The authors should consider re-wording this section to be clear which strategies might apply to the Asia Pacific. For example, it's not clear whether the authors are suggesting that YF vaccination should be mandatory in the Asia Pacific? Mass vaccination in the Asia Pacific is not a justified strategy.

5) Lines 206–207. "Therefore, combining a secured production of YFV vaccine (still produced in embryonated chicken eggs causing occasional issues of supply) and new promising vector control methods (e.g. Wolbachia strategy) would significantly improve the prevention and control of YF as of dengue."

Vector control strategies in the Asia Pacific are unlikely to be justifiable strategies for preventing potential YF outbreaks. Perhaps additional requirements for inbound travellers from YF risk areas and detection/surveillance capabilities in high-risk areas and vaccine stockpiling and manufacturing surge capacity would be consider before this.

It is also not correct to discuss the "control" of YF in the Asia Pacific where there is no transmission.

Answers to Reviewer #1

- **Point 2.1 too few data points:**

Geographic restriction of the analysis has improved this part of the analysis, but poor geographic coverage remains a key issue. There are only 10 geographically distinct points (with 4 very closely located in Taiwan) with predictions made thousands of miles (and in completely different countries) from the data included in this analysis. Any attempt at extrapolation gives, in my view, a false impression of an understanding of the determinants of geographic variation and geographic precision in prediction.

As suggested by the reviewer, we have reconsidered the way to interpret our data and made the changes requested to better fit with our data; see answer below.

- **Point 2.2 Methods for extrapolation:**

Apologies I probably should have been clearer in my original review. Inverse distance weighting imposes (i.e. assumes) a very strong unfitted functional form on spatial autocorrelation. Ideally this assumption would be supported by an empirical analysis of the spatial autocorrelation of the data e.g. a semi variogram. If this shows that there is no spatial autocorrelation the authors will need to examine regression-based approaches using covariates or remove the extrapolation part of this analysis entirely and just calculate vector competence for the field sites where they do have data.

We agree with the reviewer that two distinct and valid points have been raised here: *i)* suitability of inverse-distance weighting to the data we have collected and present as part of this work and *ii)* the accuracy of the geographic resulting of such a model, given the data serving as input. With respect to Point 2.2, we have investigated spatial autocorrelation and found that the semi-variogram, although noisy, tended to reach a plateau after ~300 km. However, we concur with Point 2.1 that 10 data points (4 of which highly clustered in space) are not sufficient to support the assumption of spatial autocorrelation. As a result, we have decided after careful consideration to discard the geographic smoothing approach from the manuscript (also removed the Fig. 6 that was used to generate the risk map).

Instead, we have decided to present the result of the mosquito transmission efficiency for YFV at the sampling points jointly with the former results from Kraemer *et al.* While this does not affect the main points of the manuscript (namely, that competent vectors can be identified in South-East Asia, where vaccine coverage remains poor), we agree that this simplified approach will address the constructive methodological caveats mentioned above. As a result of these changes, the manuscript has been expunged of the “**Modeling of geographic dissemination risk**” paragraph in the methods. Instead, we now provide a shortened paragraph (Lines 279-283):

“Risk of *Aedes aegypti*-mediated YFV transmission

The work by Kraemer *et al.*³⁰ presents worldwide estimates of the occurrence of *Ae. aegypti*, i.e. the probability of encountering *Ae. aegypti* throughout at a resolution of 5 km x 5 km. We extracted these values at the sampling points where studied mosquito populations can be found as well as averaged these at each geographical points to illustrate possible heterogeneity in mosquito occurrence.”

The corresponding result paragraph was also updated (Lines 152-159):

“Ability of different *Aedes aegypti* populations to transmit YFV

To determine the risk of mosquito-mediated YFV transmission at each location, we used transmission efficiencies (Fig. S1) and probabilities of vector occurrence (data from Kraemer et al. 2015³⁰). When considering only *Ae. aegypti* from Asia, regions where CSP (Thailand), TRUNG (Vietnam) and NANZI (Taiwan) populations are located, presented a higher transmission risk of YFV (CSP: 54% [32.8% – 74.4%], TRUNG: 25% [9.8% – 48.7%], NANZI: 21% [7.1% – 42.2%]). In these regions, mosquito occurrence is predicted to be high and overall constant within a 5 km radius, allowing for competent vectors to place immunological naive populations (humans and natural reservoirs) at risk of YFV infection (Fig. 6).”

The associated Figure 6 has also been improved to reflect these changes, and the legend updated accordingly.

- **Point 2.3 Kramer maps**

The changes are largely appropriate, although statements like “we used this quantity to reflect human exposure to mosquitoes” are still not inkeeping with interpretations of this map as “human exposure to mosquitoes” is dependent both on presence of a vector population, its abundance and human biting rate.

As per our answers to Points 2.1 and 2.2, we have decided not to combine the modeled vector occurrence values from Kraemer et al. with our estimates. We now make use of these data only to extract the occurrence values at geographical sampling points where were sampled the studied mosquito populations. The updated Figure 6 now shows two panels: A) the subset of the map by Kraemer *et al.* restricted to South-East Asia and B) the mosquito transmission efficiencies for YFV at each location.

We have substantially rephrased how to interpret the quantity from this map throughout the manuscript. Furthermore, we now make use of this map to aggregate the predicted vector occurrence probabilities at different geographic scales and present the results in Figure 6 (coloured bars in Figure 6B), along with our observed transmission efficiencies in the laboratory (grey bar in Figure 6B). There are indeed clear limitations to extending results from laboratory-based experiments to wild mosquito behavior. These limitations are now discussed more extensively in the discussion (Lines 200-222):

“Likewise, a recent modeling exercise (data extracted from³⁰) suggests that *Ae. aegypti* can commonly be found throughout South-East Asia (Fig. 6A). These results suggest that *Ae. aegypti* from the Asia-Pacific region are competent to YFV and prone to trigger a YF outbreak, strengthening the conclusions drawn from metapopulation models to assess the probabilities of YFV spread based on international airline transportation⁴², or disease transmission models using infection data, vaccination coverage, and different environmental factors^{43,44}. However, it is important to note that assessing the risk of YFV transmission based on vector competence data as was done in our study conducted in laboratory conditions, does not reflect alone the capacity of mosquitoes to act as a field vector. Some environmental factors might shorten mosquito lifespan and, therefore diminish the probability of infecting after the extrinsic incubation period. Moreover, the viral titers used in our experimental infections may differ from viremias encountered in patients, 4.98 (3.50–5.79) Log₁₀ copies/mL of YFV RNA in blood⁴⁵. Finally, viral transmission in our study was determined by detecting viral particles in mosquito saliva collected using the forced salivation technique (see Methods) which do not reflect the physiological dose of viral particles delivered by a mosquito during the bite. Moreover, vector capacity integrates biotic and abiotic factors in addition to vector competence, and therefore, varies in space and time across a region; it can be influenced by population density, vector feeding behavior, and vector lifespan⁴⁶. Apart from making vaccination mandatory,

preventing YF outbreaks in the region should rely on controlling *Ae. aegypti* populations, particularly in regions suffering from dengue, chikungunya, and Zika. Although it seems difficult considering the failure in preventing and controlling dengue using conventional insecticides, combining an early detection of imported cases, a vaccination mandatory for travellers returning from countries at risk, a plan for implementing mass vaccination campaigns and securing the vaccine stockpile (still produced in embryonated chicken eggs causing occasional issues of supply), and new promising vector control methods (e.g. *Wolbachia* strategy) would significantly improve the prevention of YF as of other arboviral diseases⁴⁷.”

- **Point 2.4 Vectorial capacity equation**

Despite renaming this measure to “vector competence” I still take issue with the definition of “vector competence” as “an overview of mosquitoes’ ability to get infected from human blood and transmit back in the general population” because such a measure would still need to include the probability of surviving the EIP and the human biting probability. As it currently stands this formula only measures the probability of a single mosquito being infected, not the probability of transmission back to a human population.

As per our answers to Points 2.1 and 2.2, the definition of the (formerly) estimated quantity should not be an issue anymore. We however thank the reviewer for raising the issue about clearly defining *vector competence* in the context of our experiments. As it was formerly presented, it is true that the estimated quantity reflected on the probability for a mosquito to infect itself upon biting and only become potent for YFV transmission conditionally on surviving the EIP. This also implicitly required an infected human to get bitten.

In the current version, we define vector competence.

- **Point 2.5 Uncertainty**

Good to see the new maps of uncertainty – maybe add 95% CIs when presenting these figures in the relevant results section?

The corresponding 95 CIs are now incorporated throughout the text.

I think more fundamentally, the authors should consider what value this part of the analysis adds to their overall study and if the assumptions they are making here detract and undermine the findings from the rest of their analysis. If “vector competence” (or a modified version of it- point 2.4) were calculated just for the sites where data were available and no extrapolation was attempted would this qualitatively affect the results of this paper? Looking at the text for this section in the results section I don’t think anything would change. I should re-iterate my support for the quality of the rest of the analysis and manuscript, but my view that the geographic extrapolation component of this work is methodologically insufficient, misleading and unnecessary to support the main conclusions of the work remains.

We are grateful for the encouraging and valid comments raised in this review. We concur that overall, the key message of the work did not necessarily require enhanced modeling methods as support. The main points we raise here is that South-East Asia populations are i) mostly unprotected against YFV, ii) likely exposed to *Ae. aegypti* mosquitoes with low local heterogeneity in occurrence and iii) that these mosquitoes are competent to serve as vectors for YFV. We hope that we have addressed the caveats that were highlighted and are pleased that the reviewer finds our work to be a valuable contribution to the current literature for that subject.

Answers to Reviewer #3

- **Point #1: In particular, probability of occurrence estimated by Kraemer et al should not be interpreted as the probability of encountering a mosquito. The probability of a human encountering a mosquito would also require information on vector abundance and human population density – and so I am not convinced that vector competence (as per the definition on line 279) has been estimated.**

Following the suggestions from Reviewer #1 and after careful consideration, we have decided not to include the geographical modeling approach in this revised version of the ms. In particular, the Point #1 raised here by the reviewer yields the issue of clearly defining “vector competence” in the (former) context of the quantity generated by combining mosquito occurrence probability and mosquito ability to infect itself upon biting an infected human.

Among the caveats raised, a limitation was that wild mosquitoes were noted as likely to survive the EIP as those laboratory-raised. Furthermore, infecting wild mosquitoes would require encounters with infected hosts and successful biting. As we did not have data to inform such parameters and in light of the statistical noise owing to the irregular geographical sampling, we have decided to remove the modeling section. Instead, we now present side by side the modeled occurrence from Kraemer *et al.* (restricted to our region of interest) as well as the observed estimates of laboratory-measured vector competence.

By discarding this modeling exercise, we hope that the revised version of the manuscript will not be over-interpreted.

- **Point #2: Please add a description of the findings from reference 27 and compare to the current study results (as done for reference 28).**

We have added the following information in Lines 170-172: “A previous study using Asian *Ae. aegypti* populations showed that in laboratory conditions, *Ae. aegypti* from Laos (Bolikhamsai province) were able to transmit YFV at least 14 days after exposure to YFV S-79 strain²⁷.”

- **Point #3: Contextualisation of results from the risk estimates is still lacking from the discussion. There are other published studies which estimate the risk of YF transmission in the Asia Pacific, please discuss your results in the context of these.**

We have added in the new version in Lines 201-205:

“These results suggest that *Ae. aegypti* from the Asia-Pacific region are competent to YFV and prone to trigger a YF outbreak, strengthening the conclusions drawn from metapopulation models to assess the probabilities of YFV spread based on international airline transportation⁴², or disease transmission models using infection data, vaccination coverage, and different environmental factors^{43,44}.”

- **Point #4: The new limitations section only discusses limitations related to the transmission risk model. Limitations of the laboratory experiments and analyses should also be discussed.**

We have included in the revised version in Lines 205-212:

“However, it is important to note that assessing the risk of YFV transmission based on vector competence data as was done in our study conducted in laboratory conditions, does not reflect alone the capacity of mosquitoes to act as a field vector. Some environmental factors might shorten mosquito lifespan and, therefore diminish the probability of infecting after the extrinsic incubation period. Moreover, the viral titers used in our experimental infections may differ from viremias encountered in patients, 4.98 (3.50–5.79) Log₁₀ copies/mL of YFV RNA in blood⁴⁵. Finally, viral transmission in our study was determined by detecting viral particles in mosquito saliva collected using the forced salivation technique (see Methods) which do not reflect the physiological dose of viral particles delivered by a mosquito during the bite.”

- **Point #5: Lines 204-206. “Apart from making vaccination mandatory, preventing YF outbreaks in the region should rely on controlling *Ae. aegypti* populations, which seems difficult considering the failure in preventing and controlling dengue using conventional insecticides.**

The authors should consider re-wording this section to be clear which strategies might apply to the Asia Pacific. For example, it’s not clear whether the authors are suggesting that YF vaccination should be mandatory in the Asia Pacific? Mass vaccination in the Asia Pacific is not a justified strategy.

Lines 206–207. “Therefore, combining a secured production of YFV vaccine (still produced in embryonated chicken eggs causing occasional issues of supply) and new promising vector control methods (e.g. *Wolbachia* strategy) would significantly improve the prevention and control of YF as of dengue.”

Vector control strategies in the Asia Pacific are unlikely to be justifiable strategies for preventing potential YF outbreaks. Perhaps additional requirements for inbound travellers from YF risk areas and detection/surveillance capabilities in high-risk areas and vaccine stockpiling and manufacturing surge capacity would be consider before this.

It is also not correct to discuss the “control” of YF in the Asia Pacific where there is no transmission.

We have added in Lines 218-222:

“combining an early detection of imported cases, a vaccination mandatory for travellers returning from countries at risk, a plan for implementing mass vaccination campaigns and securing the vaccine stockpile (still produced in embryonated chicken eggs causing occasional issues of supply), and new promising vector control methods (e.g. *Wolbachia* strategy) would significantly improve the prevention of YF as of other arboviral diseases⁴⁷.”

REVIEWERS' COMMENTS

Reviewer #1 (Remarks to the Author):

The reviewers have sufficiently addressed all my comments.